# Semi-supervised integration of single-cell transcriptomics data

Massimo Andreatta [1,2,3], Léonard Hérault [1,2,3], Paul Gueguen [1,2,3], David Gfeller [1,2,3], Ariel J. Berenstein [4] & Santiago J. Carmona [1,2,3] ✉

Batch effects in single-cell RNA-seq data pose a significant challenge for comparative analyses across samples, individuals, and conditions. Although batch effect correction methods are routinely applied, data integration often leads to overcorrection and can result in the loss of biological variability. In this work we present STACAS, a batch correction method for scRNA-seq that leverages prior knowledge on cell types to preserve biological variability upon integration. Through an open-source benchmark, we show that semi-supervised STACAS outperforms state-of-the-art unsupervised methods, as well as supervised methods such as scANVI and scGen. STACAS scales well to large datasets and is robust to incomplete and imprecise input cell type labels, which are commonly encountered in real-life integration tasks. We argue that the incorporation of prior cell type information should be a common practice in single-cell data integration, and we provide a flexible framework for semi-supervised batch effect correction.

Single-cell omics technologies enable characterizing the cellular complexity of biological samples with very high resolution. While individual samples can provide readouts for thousands of individual cells, addressing biological questions typically requires the comparative analysis of multiple samples, tissues, individuals and experimental conditions. By means of data integration or harmonization, cells from different sources can be placed in the same embedding or latent space, facilitating the measurement of distances between them, the collective annotation of cell populations, and additional downstream joint analyses. However, differences in sample collection, processing, and experimental protocols introduce unwanted variation in the molecular readouts that interferes with the identification of true biological differences between samples. This technical variation is sometimes referred to as "batch effects" since it is typically observed between groups of samples that were handled in different batches[1,2].

Several methods for single-cell RNA-seq data integration have been proposed, based on different approaches such as mutual nearest neighbors and linear embeddings, deep learning, and graph structures, each with strengths and limitations[3–8]. Integration methods aim at removing batch effects while preserving relevant biological variation. Two main aspects are considered to determine the quality of single-cell data integration: (i) batch mixing and (ii) preservation of biological variance. Batch mixing measures whether similar cells from different batches are well mixed after integration. Frequently used metrics of batch mixing are entropy, kBET, and integration LISI (iLISI)[9–11]. Preservation of biological variance can be quantified by how close to each other cells of the same type are, and how separated from each other cells of different types are in the joint integrated embeddings. Commonly-used metrics include average silhouette width (ASW), average Rand index (ARI), and cluster LISI (cLISI)[11–13]. For a review on integration metrics see Luecken et al.[14].

For certain tasks, such as integration of technical replicates or very similar samples, most integration methods perform generally well both in terms of batch mixing and preservation of biological variance[2,14]. However, more common scenarios include the integration of datasets from biologically heterogeneous samples, e.g. from different donors, timepoints or tissues. These do not only display technical batch effects, but also large variability in terms of cell type

---

[1]Department of Oncology, Lausanne Branch, Ludwig Institute for Cancer Research, CHUV and University of Lausanne, 1011 Lausanne, Switzerland. [2]AGORA Cancer Research Center, 1005 Lausanne, Switzerland. [3]Swiss Institute of Bioinformatics, 1015 Lausanne, Switzerland. [4]Laboratorio de Biología Molecular, División Patología, Instituto Multidisciplinario de Investigaciones en Patologías Pediátricas (IMIPP), CONICET-GCBA, Buenos Aires C1425EFD, Argentina. ✉e-mail: santiago.carmona@unil.ch

composition. Differences in cell type abundance are a major component of biological variance across samples[15,16], and such imbalance represents a particularly challenging task for integration methods[17,18].

The choice of a specific integration method and parameters configuration should take into account the tradeoff between preserving relevant biological variation and increasing batch mixing[14]. Since most integration tasks involve samples with some degree of cell type imbalance, choosing integration strategies that favor preservation of biological variance over batch mixing might be preferable[17]. An appealing approach to preserve biological variance is to make use of prior cell type information to guide dataset integration. Indeed, in recent benchmarks of single-cell data integration tools, methods that take cell type labels as input showed the highest performance in terms of preservation of biological variance[14,19].

In this study, we describe STACAS, a semi-supervised scRNA-seq data integration method that leverages prior knowledge in the form of cell type annotations to preserve biological variance during integration. Using an open and reproducible benchmarking pipeline we show that semi-supervised STACAS compares favorably to popular unsupervised methods such as Harmony, FastMNN, Seurat v4, scVI, and Scanorama, as well as to the supervised methods scANVI and scGen, while being robust to missing and imperfect cell type information. We argue that prior cell type information should be routinely incorporated in integration tasks and we propose a general strategy for its implementation.

## Results

### Semi-supervised STACAS uses prior cell type information to guide data integration

STACAS is a batch correction method for the integration of heterogeneous scRNA-seq datasets. The result of STACAS integration is a batch-corrected combined gene expression matrix that can be used for downstream multi-sample analyses, such as clustering and visualization. The method is based on the concept of mutual nearest neighbors to identify biologically equivalent cells in pairs of datasets (referred to as "anchors")[3], which are used to estimate batch effects.

STACAS builds upon the Seurat integration method[5] and applies reciprocal principal component analysis (rPCA) to find anchors, where each dataset in a pair is projected into the principal components (PC) space of the other. STACAS uses the rPCA distance between the two cells of an anchor to weigh the biological relevance of the anchor and ultimately its contribution to batch correction vectors (see Methods). In this way, anchor cells that are close to each other in the rPCA space will contribute more strongly to batch correction than distant anchor cells, which are transcriptionally more dissimilar and thus less likely to be biologically equivalent cells.

In anchor-based methods, obtaining an accurate set of anchors is critical for integration performance. STACAS v2 introduces the ability to use prior information, in terms of cell type labels, to refine the anchor set. We refer to this mode of dataset integration as "semi-supervised". Cell type labels may be obtained from automated classifiers, manual annotation, multi-modal information or any other source. When provided, cell labels are used by STACAS to remove "inconsistent" anchors, composed of cells with different labels (Fig. 1). Note that missing labels are not penalized in this step, generalizing the method to partially annotated datasets. Finally, consistent integration anchors are used to calculate batch effect correction vectors between pairs of datasets, and their weighted anchor scores are used to construct an integration guide tree that will determine the order in which the datasets will be integrated (see Methods).

### A cell type-aware implementation of the LISI metric to quantify batch mixing

A frequently used metric to assess the quality of single-cell data integration is the Local Inverse Simpson's Index (LISI). LISI measures mixing by estimating the effective number of classes in local neighborhoods of cells[11] and is relatively fast to compute. When applied to measure the effective number of datasets or batches in a neighborhood, this metric is referred to as 'iLISI' (integration LISI); when applied to measure the effective number of cell types in a neighborhood, LISI has been named 'cLISI' ("cluster" or "cell type" LISI). Another widely-used performance metric to assess cell type

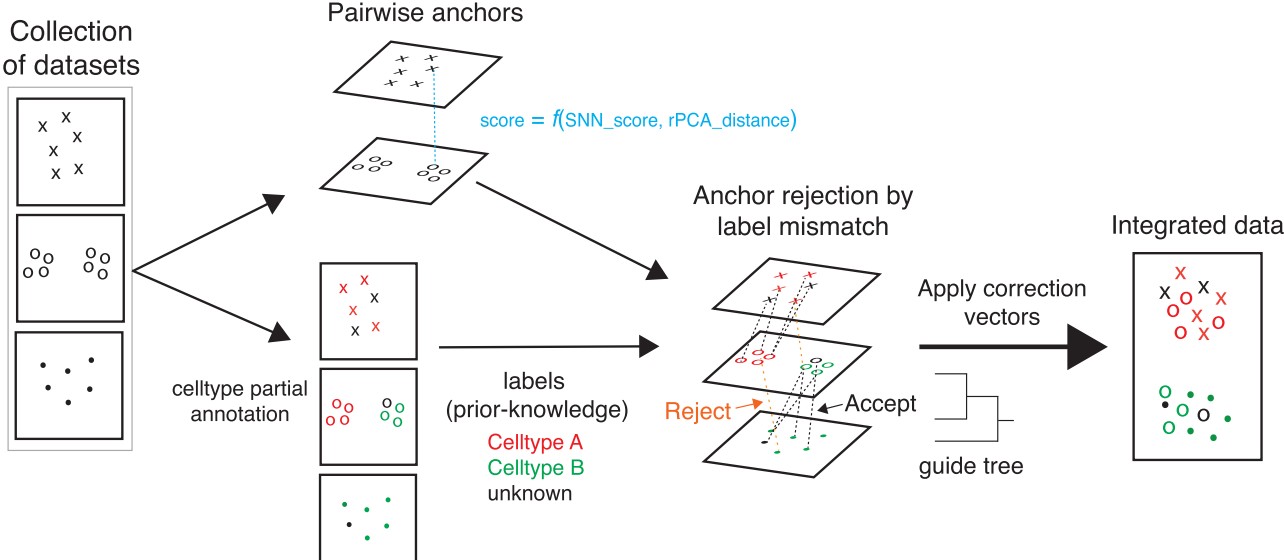

**Fig. 1 | Schematic of semi-supervised STACAS integration method.** The algorithm identifies integration anchors between all pairs of datasets from a shared nearest neighbors (SNN) graph. These are expected to be cells of the same type across batches and are used to calculate batch effects. Integration anchors are weighted by a score that combines a SNN anchor consistency score (based on the overlap of shared neighbors) and a score based on rPCA distance (how similar are cells of one dataset to the corresponding anchor cells in a second dataset projected into the PCA space of the latter). If cell type labels are available, they can be provided as input to the algorithm. When cell type labels between two cells of an anchor are inconsistent, the anchor is rejected with a predefined probability and in that case will not contribute to batch effect correction. Finally, the sum of retained, weighted integration anchor scores is used to calculate global similarities between datasets and to derive a guide tree that will determine the order in which the datasets are to be integrated.

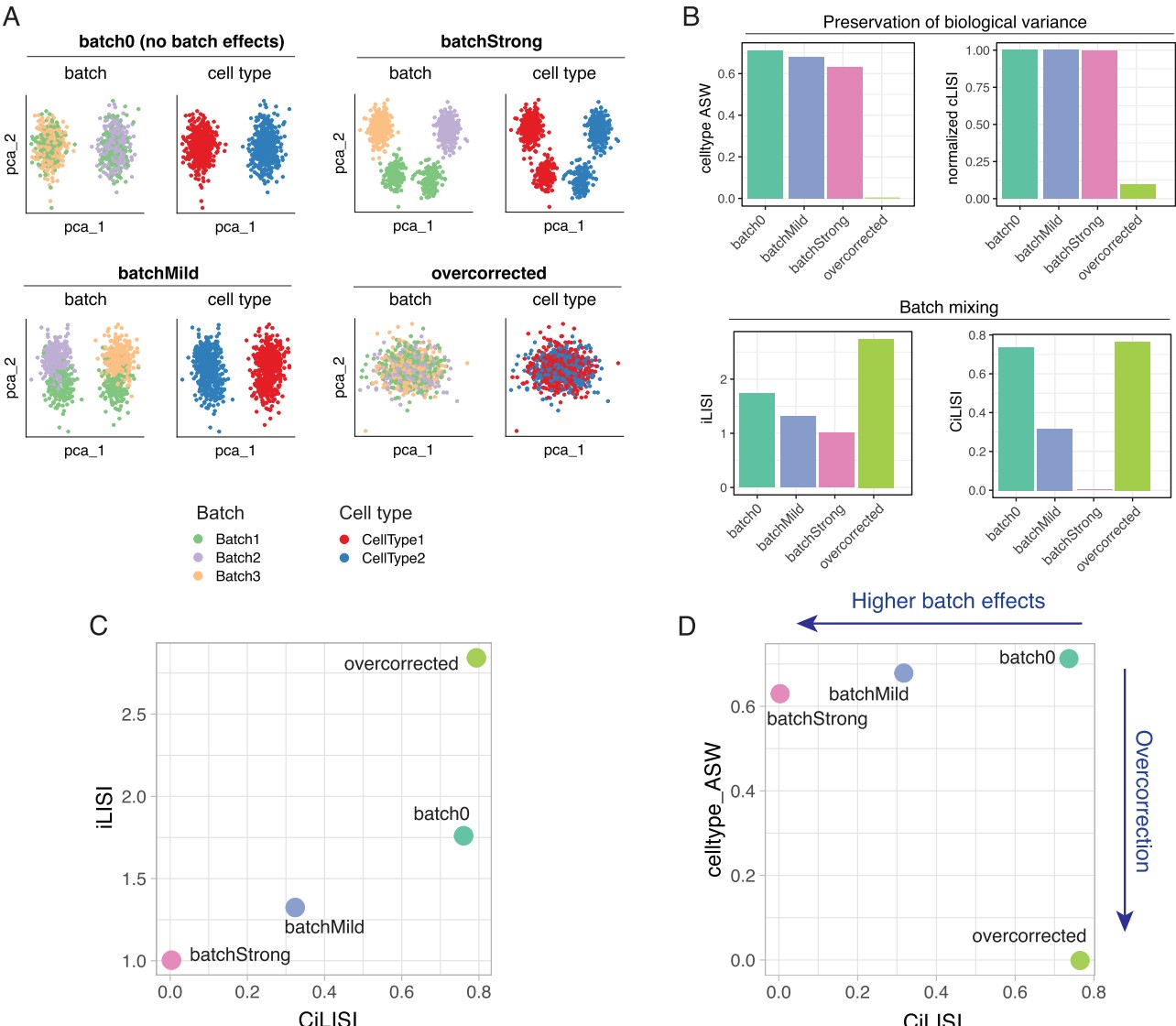

**Fig. 2 | Integration metrics on synthetic single-cell datasets. A** Scatterplot of first and second principal components (pca_1 and pca_2) for four simulated scenarios where datasets have increasing levels of batch effects ("batch0" with no batch effects, "batchMild", and "batchStrong"), and one case where there is no batch effect and no biological cell type signal (simulating the result of an extreme batch-effect "overcorrection"). Each scenario is composed of three samples/batches and two cell types. **B** Metrics of preservation of biological cell type variability (cell type ASW and average normalized cLISI) and batch mixing (average iLISI and average CiLISI) for the four simulated scenarios. **C** Scatterplot of average CiLISI (x-axis) versus average iLISI (y-axis) for the four simulated scenarios. **D** Scatterplot of average CiLISI (x-axis) versus average cell type ASW (y-axis) for the four simulated scenarios; a batch-corrected and biological variance-preserving integration should aim at maximizing both metrics. CiLISI is defined here as the normalized batch LISI calculated on a per-cell type basis; it tends to 1 when cells of the same type are well mixed across batches, and becomes zero when cells from different batches do not mix. AWS average silhouette width, LISI local inverse Simpson's index, iLISI integration (or batch) LISI, cLISI celltype (or cluster) LISI, CiLISI per celltype integration LISI. Source data are provided as a Source Data file.

clustering is the average silhouette width (ASW), which quantifies distances of cells of the same type compared to the distances to cells of other types (cell type ASW)[2,14].

To evaluate the behavior of these metrics on datasets with different levels of batch effects, we generated synthetic scRNA-seq datasets (see Methods). This simulation setting consists of three biological samples: sample A comprises cell type 1 and cell type 2 in equal parts; sample B contains only cell type 2; and sample C contains only cell type 1. Each sample corresponds to a different experimental batch (batch 1 to 3) and displays batch effects in addition to cell type biological variation (Fig. 2A). We generated several scenarios with increasing batch effects: no batch effects (batch0), mild (batchMild), and strong batch effects (batchStrong). In the fourth example both cell type variation and batch effects are zero, representing the situation of

an extreme overcorrection of batch effects. We employed these simulated datasets to evaluate different integration metrics.

In terms of preservation of biological variance, both normalized cell type ASW and normalized cLISI (see methods) correctly capture the poor cell type separation in the 'overcorrected' dataset, while remaining high in all other cases (Fig. 2B). We note that the normalized cLISI, because it only measures local neighborhoods, is unaffected by mild levels of batch effect. Instead, cell type ASW seems to be more sensitive in detecting cell type spread due to batch effects (Fig. 2B). In terms of batch mixing, iLISI decreases together with the mixing of cells from different batches, as expected. However, iLISI increases from ≈1.75 in the case with no batch effect (batch0) to ≈2.75 in the case with no batch effect and no cell type signal (overcorrected) (Fig. 2B). Hence, iLISI would favor a method that completely removes biological

variance together with batch effects over a method that effectively removes batch effects while preserving biological variance. This is an undesirable behavior for an integration metric. To obviate this limitation, we propose to evaluate iLISI on a per-celltype basis, hereafter referred to as CiLISI (normalized to vary between 0 and 1, see Methods). Unlike iLISI, CiLISI measures batch mixing in a cell type-aware manner and scores similarly the cases with no batch effects, irrespective of the biological variance (Fig. 2C). We argue that CiLISI is preferable over iLISI because it does not penalize methods that preserve biological variance in datasets with cell type imbalance.

Given these insights, we suggest evaluating integration performance by assessing jointly (i) batch mixing in terms of per-cell type normalized batch LISI (CiLISI) and (ii) cell type clustering in terms of cell type ASW (or alternatively normalized cLISI). Well-performing methods should be able to mix cells of the same type in different batches (i.e. maximize CiLISI) while keeping apart cells of different types (i.e. maximizing cell type ASW and normalized cLISI) (Fig. 2D). We implemented these metrics in an R package, available at https://github.com/carmonalab/scIntegrationMetrics.

## Semi-supervised STACAS outperforms state-of-the-art methods

To assess the performance of semi-supervised STACAS compared to state-of-the-art integration tools, we took advantage of the 'scib' pipeline published by Luecken et al. [14]., which allows comparing methods written in R and python in a reproducible environment. We modified the pipeline to include STACAS and the CiLISI metric, and to evaluate all methods on latent spaces of the same size (see Methods for details). Importantly, we included the option to evaluate semi-supervised methods with incomplete cell type labels, as well as with partially shuffled labels. In our benchmark we shuffled 20% of the cell type labels and set 15% to 'unknown' as input to supervised methods, simulating a realistic setting where prior cell type knowledge is imperfect and incomplete. We assess the effect of overfitting in different integration methods as a function of the percentage of incomplete or noisy input labels in a later section.

The performance of 11 computational tools was evaluated on 4 different integration tasks, with increasing levels of cell type imbalance (Fig. 3A–D). We also evaluated the effect of re-scaling gene expression data prior to batch effect correction, and we report results for both unscaled and scaled data for each method (this is not applicable to scVI and scANVI, which use count data as input). On the Pancreas integration task, consisting of technical replicates obtained with different sequencing platforms, most methods performed well (Fig. 3A). In particular, Seurat CCA, STACAS (unsupervised) and Seurat rPCA achieved high batch mixing (CiLISI) while preserving biological variance (cell type ASW). In this balanced scenario (Fig. S1A), cell type information does not appear to provide an advantage to semi-supervised methods: semi-supervised STACAS (ssSTACAS) obtained similar performance to unsupervised STACAS, scGen ranked below several unsupervised methods, and scANVI performed poorly and only marginally better than its unsupervised counterpart scVI.

The Immune integration task comprises 10 datasets from 5 different studies corresponding to peripheral blood and bone marrow human samples, as compiled by Luecken et al. [14]. While most cell types are represented by multiple datasets, the composition is less balanced than in the Pancreas integration task (Fig. S1B) Here the semi-supervised methods scGen and ssSTACAS were the best tools in finding a trade-off between batch mixing and preservation of biological variance (Fig. 3B). Other methods such as Seurat CCA, scVI and Harmony, on the other hand, overcorrected batch effects and performed poorly in terms of cell type ASW (cfr. Figure 3B and 'overcorrected' in Fig. 2D).

The Lung atlas represents a more challenging integration task, as it comprises samples from multiple human donors, covering different spatial locations [20]. Semi-supervised STACAS, and to a lesser degree its

unsupervised version, successfully mitigated batch effects while preserving biological variance (Fig. 3C). Again, the widely used Seurat CCA, Harmony and scVI performed poorly in terms of cell type AWS, suggesting that these methods tend to overcorrect batch effects.

To evaluate the performance of STACAS in a setting with strong cell type imbalance and strong batch effects, we used the collection of T cell datasets from Andreatta et al. [21]. These contain seven datasets from six different studies covering tumor and lymph node samples, comprising studies with both CD4+ and CD8+ T cells (MC38_dLN, Ekiz and Xiong), only CD8+ T cells (Carmona, Singer) or only CD4+ T cells (Magen_dLN and Magen_TILs) (Fig. S1D). As with the Immune integration task, we observed that semi-supervised STACAS and scGen were the best performing tools, especially in terms of preservation of biological variance (Fig. 3D–F, Fig. S2).

Globally, when considering cell type ASW and CiLISI across the four integration tasks, semi-supervised STACAS was the best-performing method, followed by unsupervised STACAS and scGen (Fig. S3B–D). scANVI performed in the middle of the pack in terms of these metrics, ranking 8th/9th globally (Fig. S3B–D). When evaluating performance on a broad panel of metrics for preservation of biological variance ("bio-conservation") and batch-correction, as in the original benchmark by Luecken et al., semi-supervised STACAS remains the best method, followed by the semi-supervised method scANVI (Fig. 3G, S3A, C). These rankings remained consistent whether the latent spaces used 30 or 50 dimensions (Fig. S3, S4). Across all integration tasks, using unscaled data was preferable to scaled data for the preservation of biological variance in terms of cell type silhouette coefficient (Fig. S5). In integration tasks with large cell type imbalance, methods that use prior cell type information were shown to better preserve biological variance compared to unsupervised methods. To evaluate more systematically the effect of cell type imbalance on integration performance, we artificially generated modified versions of the pancreas dataset with increasing levels of cell type imbalance. In these settings, ssSTACAS consistently outperformed the other methods across a wide range of cell type imbalance (Fig. S6). In particular, when evaluating rankings based on CiLISI and celltype_ASW, ssSTACAS was the top-performing method in all cases except the fully balanced set. The other supervised methods scANVI and scGen also showed a relative increase in ranking as cell type imbalance increased. In contrast, Seurat CCA was the top performer with balanced cell-type composition, but its performance dropped sharply as soon as imbalance was added to the data (Fig. S6A). Similar patterns could be observed when evaluating performance on a larger panel of metrics (Fig. S6B). Based on these results, we recommend using prior cell type information, whenever available, to guide single-cell data integration tasks.

## Semi-supervised STACAS is robust to incomplete and noisy annotations

A potential risk of applying supervised or semi-supervised integration methods is overfitting the cell type labels provided as input. In real-life data integration scenarios, manual or automatic cell annotation may not allow providing cell type identities to all cells. Even when all cells can be annotated, these annotations might be wrong or inaccurate. Therefore, relying excessively on a priori cell type labels to force integration results may be undesirable. Instead, robust (semi) supervised integration methods should be tolerant to incomplete and incorrect input cell type information.

To evaluate the effect of incomplete or noisy cell type annotations on the performance of semi-supervised integration methods, we constructed alternative versions of the four collections of datasets used in our benchmark (Pancreas, Immune, Lung and T cell tasks) with increasing levels of shuffled or unknown cell type labels (see Methods). We used these noisy labels as input for (semi) supervised integration, and the original labels to evaluate performance. As the percentage of shuffled labels increases from 0% to 100%, we observed an expected

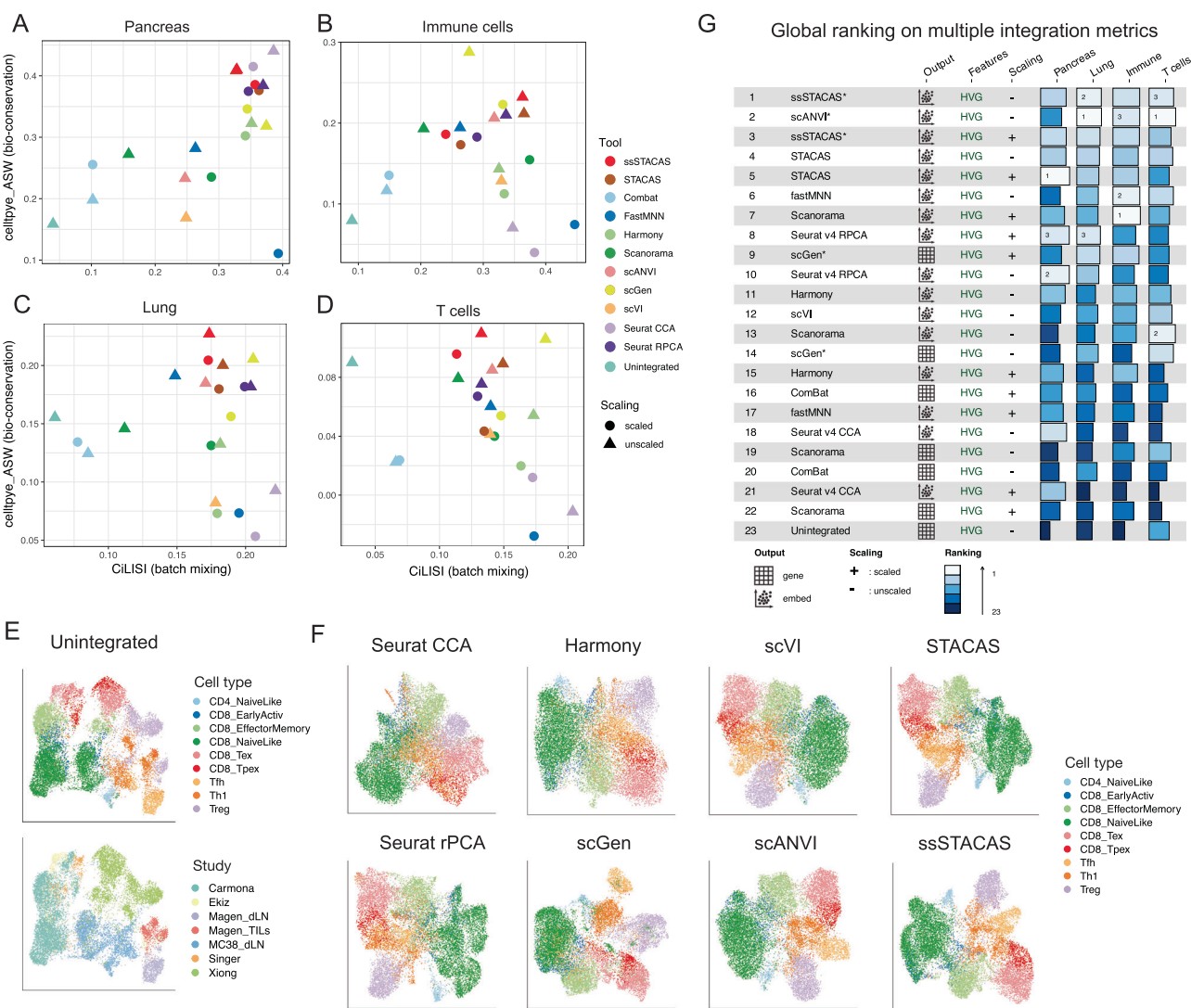

**Fig. 3 | Integration performance for single-cell data integration tools over 4 different tasks. A** CiLISI (per cell type integration LISI, measuring cell type-aware batch mixing) vs. celltype_ASW (cell type average silhouette width, measuring preservation of biological variance) for several integration methods on the Pancreas integration task. **B** CiLISI vs. celltype_ASW across methods on the Immune cells integration task. **C** CiLISI vs. celltype_ASW across methods on the Lung integration task. **D** CiLISI vs. celltype_ASW across methods on the T cells integration task. **E,F** UMAP embeddings for the mouse T cell integration task, for unintegrated data colored by cell type (top) and by study of origin (bottom) **(E)** and for eight representative integration methods, colored by cell type **(F)**. **G** Global rankings of integration tools based on the weighted contribution of a broad panel of metrics both for preservation of biological variance ("bio-conservation") and batch-correction, as proposed by Luecken et al. Supervised methods (ssSTACAS, scGen and scANVI) were provided noisy input labels (15% unknown and 20% shuffled labels). All experiments were performed with latent spaces of 50 dimensions. For alternative rankings using only CiLISI and celltype_ASW, or a different number of dimensions, see Fig. S3 and S4. HVG: highly variable genes. Source data are provided as a Source Data file.

gradual drop in performance for all supervised methods, both in terms of bio-conservation and batch mixing (Fig. 4A). However, scGen was considerably more sensitive to shuffled labels than ssSTACAS and scANVI, with a sharp drop in performance with as low as 10% or 20% of shuffled labels. Both ssSTACAS and scANVI were robust to noisy labels, but ssSTACAS obtained higher celltype ASW across all tasks. Similar results were observed when cell type label shuffling only affected neighboring, transcriptionally similar cell types (Fig. S7) or when providing incomplete cell type labels as input to the three tools (0% to 100% 'unknown' labels) (Fig. 4B). These results suggest that scGen, when used as a batch-correction tool, is prone to overfitting on the provided input labels. In general, using the same cell type labels as input for supervised integration and to evaluate integration leads to over-optimistic performance assessment and should be avoided.

These considerations suggest that fair performance evaluation of semi-supervised tools should account for potential noise in the input labels. As previously mentioned, the benchmark presented in Fig. 3 included 20% shuffled labels and 15% unknown labels. Because random shuffling of labels can, with some probability, swap identical labels, a 20% random shuffling results in practice in about 15% of actual shuffled labels of a different identity. This setup corresponds to an integration task where about 70% of cells are correctly annotated, 15% are wrongly annotated, and 15% are left unannotated. We believe this is a more reasonable setting for benchmarking than assuming the totality of true cell types can be known prior to integration. A modified version of the 'scib' pipeline that can account for incomplete or unknown annotations is available at https://github.com/carmonalab/scib-pipeline.

## Construction of a multi-study reference single-cell transcriptional map for human CD8 T cells
To date, most single-cell studies define cell states from scratch by dimensionality reduction, cell clustering and annotation. This approach

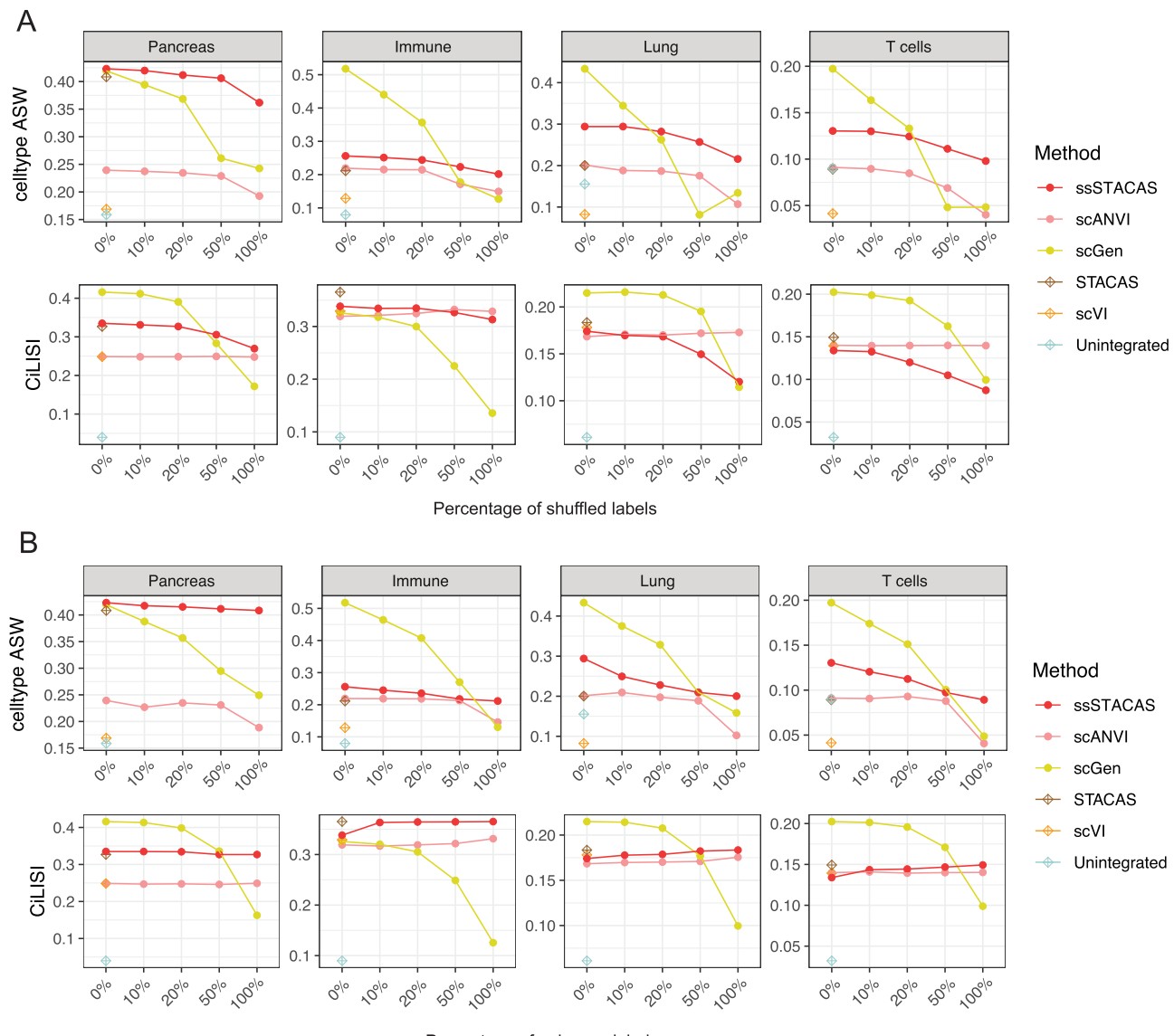

**Fig. 4 | Effect of noisy or incomplete cell type annotations on data integration by supervised or semi-supervised methods. A** Preservation of biological variance (measured by celltype_ASW) and batch mixing (measured by CiLISI) for 4 data integration tasks, using as input all cell type labels (0%) or increasing levels of shuffled cell type labels (10% to 100%). **B** Preservation of biological variance (measured by celltype_ASW) and batch mixing (measured by CiLISI) for 4 data integration tasks, using as input all cell type labels (0%) or increasing fractions of unknown cell type labels (10% to 100%). Unsupervised versions of ssSTACAS and scANVI (STACAS and scVI respectively) are included for reference. Source data are provided as a Source Data file.

is highly time-consuming and leads to inconsistent definitions across studies. Instead, the use of expert-curated reference maps to interpret single-cell data enables more consistent and faster cell state definitions[21–23]. Building robust reference single-cell maps typically requires integrating multiple datasets from different studies and conditions.

To showcase the benefits of semi-supervised STACAS to generate a multi-study reference map, we applied it to integrate multiple human CD8+ T cell single-cell datasets. Starting from a publicly available collection of tumor-infiltrating lymphocyte scRNA-seq datasets ('Utility' collection), we identified 20 high-quality samples with a sufficiently large number of cells and high subtype diversity (see Methods). These samples amounted to 11,021 cells covering 7 different tumor types from multiple studies[24–32]. Before dataset integration, we observed large batch effects between samples (Fig. 5A), as could also be quantified by a low CiLISI value (Fig. 5B). To leverage prior knowledge on CD8 T cell diversity, we defined scGate[33] gating models for six

CD8 T cell subtypes based on well-established marker genes from literature (see Methods). This model allowed annotating individual datasets with partial labels and provided prior knowledge for semi-supervised integration (Fig. 5A).

Upon semi-supervised STACAS integration, cells of the same type were clustered together, with simultaneous mitigation of batch effects (Fig. 5A, B, Fig. S8A). Compared to the uncorrected data, both batch mixing (CiLISI) and biological variance preservation (celltype ASW) were improved (Fig. 5B). Importantly, semi-supervised integration allowed higher celltype ASW compared to unsupervised STACAS integration, without a negative impact on batch mixing (Fig. 5B). Integration of these data by Harmony[11], one of the most widely used integration methods to date, resulted in high batch mixing but dramatic loss of biological variance (Fig. 5B, Fig. S8A), in agreement with the results from our benchmark. Results for additional methods are shown in Fig. S8B. On the integrated space produced by this first integration (Semi-supervised STACAS (1)), we

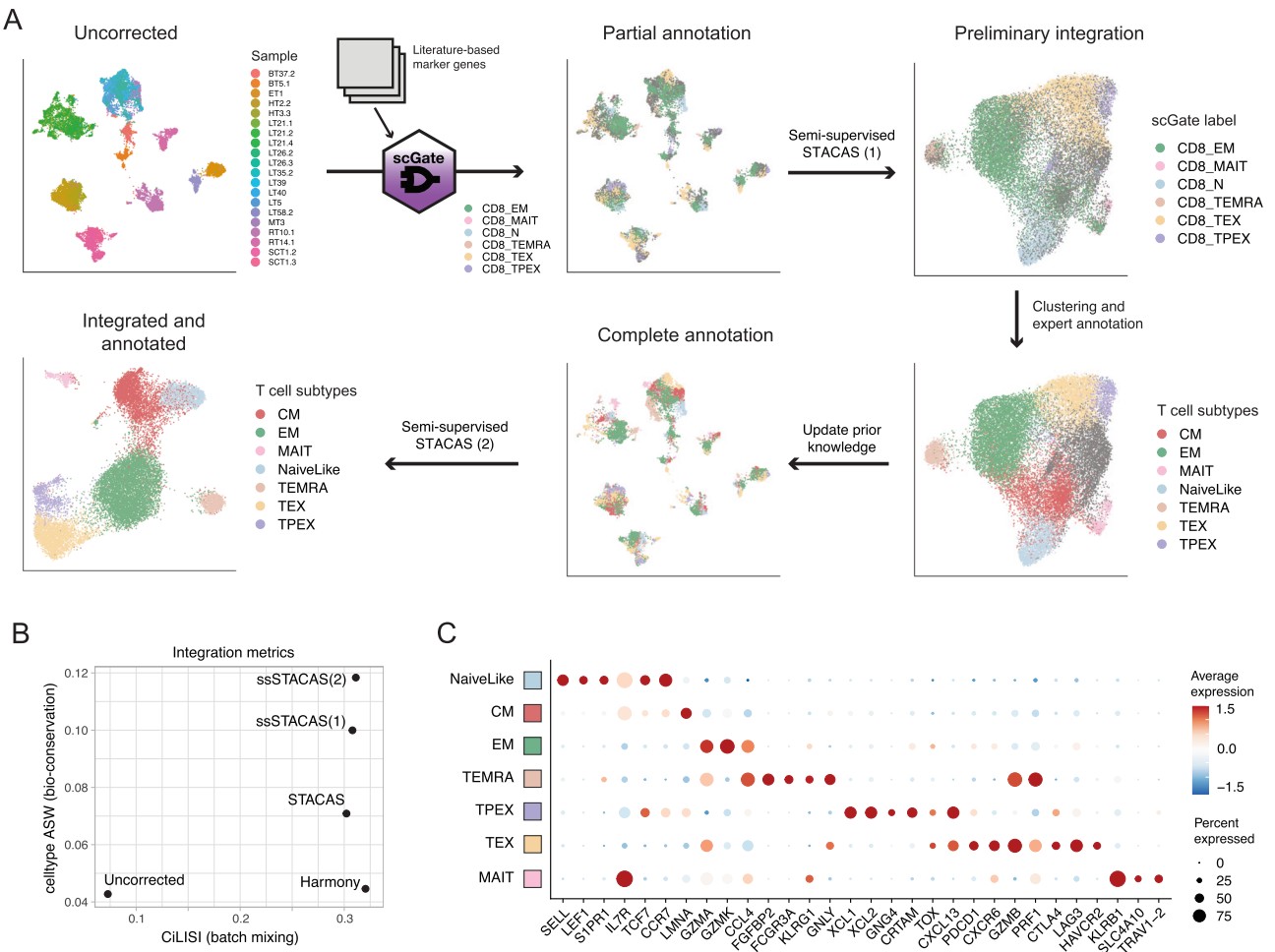

**Fig. 5 | Construction of a multi-study reference map for human CD8 T cells with semi-supervised STACAS. A** Starting from 20 samples with large batch effects, partial cell type annotations were generated using the scGate package and literature-based marker genes. These labels were used as input to a first semi-supervised STACAS integration, allowing the mitigation of batch effect and the expert annotation of clusters corresponding to T cell subtypes. These new labels were used to update the prior knowledge of the original data, and used as input for a second semi-supervised STACAS integration, to define the final integrated space of CD8 T cell subtypes. **B** Integration metrics for batch mixing (CiLISI) and biological variance preservation (celltype_ASW) for the uncorrected data, Harmony,

unsupervised STACAS, semi-supervised STACAS on the initial partial annotation by scGate [ssSTACAS (1)] and semi-supervised STACAS on the updated annotations derived from the first integration [ssSTACAS (2)]. Additional methods were compared in Figure S8B. **C** Average gene expression profiles (scaled by standard deviation) for a panel of marker genes on the seven T cell subtypes of the final integrated CD8 T cell map. Cell type abbreviations: CM central memory, EM effector memory, TEMRA terminally-differentiated effector cells (aka effector memory cells re-expressing CD45RA), TEX exhausted effector cells, TPEX precursor of exhausted cells, MAIT Mucosal-associated invariant T cells. Source data are provided as a Source Data file.

performed clustering and manually annotated the main T cell subtypes. We then went back to the original data with this updated, complete set of cell type annotations, and performed a new semi-supervised integration (Semi-supervised STACAS (2)) guided by the updated prior knowledge that was garnered from the first round of integration. The second integration allowed further improving cell-type ASW while conserving good batch mixing (Fig. 5A, B, Fig. S8A). This suggests a strategy to iteratively update and improve cell type annotations based first on prior knowledge, and secondly on the results of preliminary analyses.

The integrated single-cell map recapitulated with high resolution the known diversity of CD8 tumor-infiltrating T cells, as represented by seven different subtypes (Fig. 5C and Fig. S8C): Naive-like cells, characterized by the expression of transcription factors *TCF7* and *LEF1* and homing molecules *SELL, S1PR1* and *CCR7*; transcriptionally-related Central-memory (CM) cells, with high expression of *IL7R* and lower expression of other naive cells markers; Effector-memory (EM) cells, characterized by highest *GZMK* expression (which in the tumoral context have been referred to as 'pre-dysfunctional'[34]); terminally

differentiated effector cells (TEMRA), with high expression of *GNLY, PRF1*, multiple granzymes, *KLRG1* and *FCGR3A* (encoding CD16);[35] mucosal-associated invariant T cells (MAIT), characterized by the semi-invariant TCR chain *TRAV1-2* and expression of *KLRB1*;[36] terminally exhausted effector (TEX) T cells, expressing cytotoxic molecules (*GZMB, PRF1*), multiple inhibitory receptors (e.g. *PDCD1, LAG3, CTLA4, HAVCR2*) and the exhaustion regulator *TOX*;[37,38] and the elusive precursors of exhausted (TPEX) T cells, with co-expression of *TCF7, TOX* and *PDCD1*, and specific expression of chemokines *XCL1* and *XCL2*[21,39,40]. These T cell subtypes showed variable frequency between samples (Fig. S1E), but displayed consistent expression profiles across studies, patients, and cancer types (Fig. S8C).

Finally, we evaluated whether STACAS could be used for large-scale integration of hundreds of samples and hundreds of thousands of cells. For large-scale integration (by default, more than 20 datasets), STACAS switches to a sequential integration strategy. Because sequential integration only requires calculation of integration anchors between one dataset at a time against a reference dataset, it is relatively undemanding in terms of computational resources.

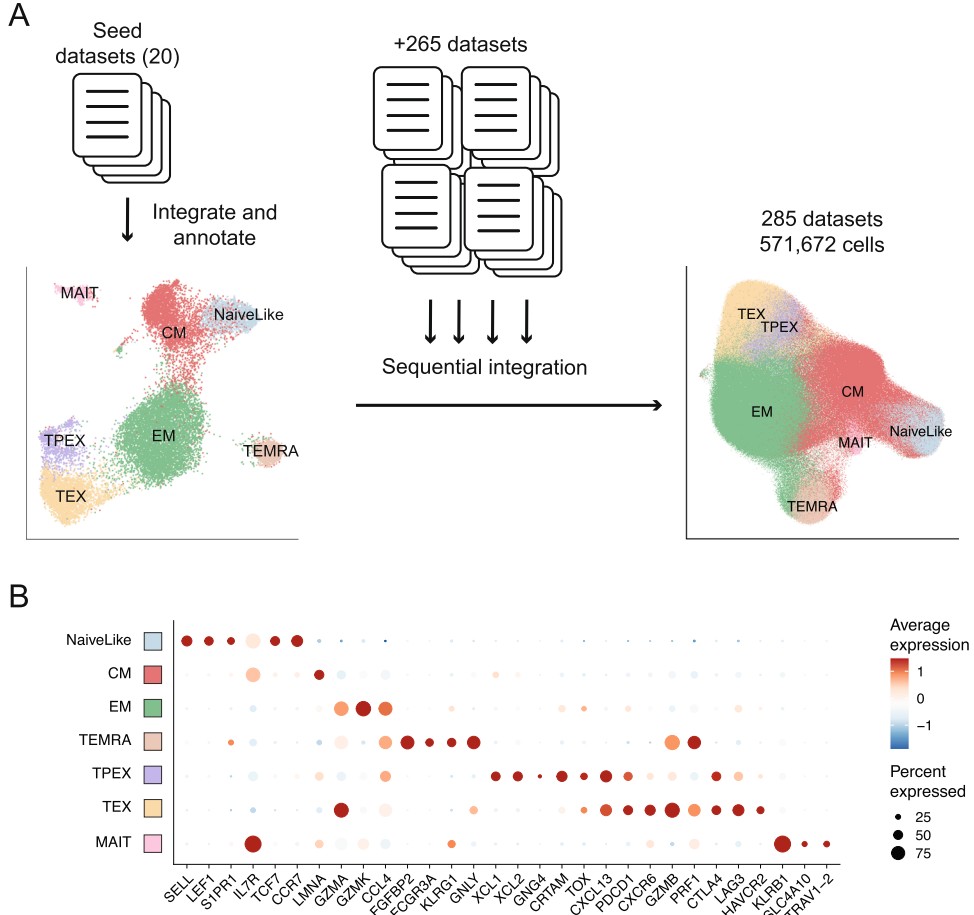

**Fig. 6 | Large scale integration of hundreds of samples. A** A set of 20 high quality "seed" datasets was selected to perform an initial integration and annotation of cell types. All remaining datasets (265 datasets) were integrated using STACAS sequential integration mode over the "seed" integrated map. Cell labels for the new, unannotated cells were obtained by a kNN classifier that assigns the majority label from the top 20 nearest neighbors from the "seed" datasets. This results in an integrated set of >500,000 cells from 285 samples. **B** Average gene expression profile for a panel of marker genes, using all cells from 285 samples. Results obtained using LIGER online learning integration are shown in Figure S9. Source data are provided as a Source Data file.

All remaining datasets from the 'Utility' collection (265 datasets, totaling 553,077 cells) were sequentially integrated on the annotated reference of "seed" datasets (Fig. 6A). This operation could be completed in approximately 150 min on a desktop computer with 64GB of RAM. After recalculation of low-dimensional embeddings, we assigned cell type labels to all new cells by K-nearest neighbor classification based on the "seed" annotated cells. This resulted in an integrated collection of 571,672 cells from 285 datasets, with average expression profiles for key T cell markers that matched expected profiles for the CD8 T cell subtypes (Fig. 6B). Cell clusters obtained upon integration by online LIGER, a tool designed for large scale integration, failed to capture the diversity of CD8 T subtypes (Figure S9). The integrated reference map, which can be directly applied to annotate additional datasets using ProjecTILs[21], is available from figshare (https://doi.org/10.6084/m9.figshare.23608308) and can be explored interactively in SPICA[41] at https://spica.unil.ch/refs/CD8T_human.

## Discussion

Integration of multiple single-cell transcriptomics datasets is a powerful approach to characterize cell diversity across tissues and conditions. Although decades of research have contributed extensive knowledge on cell markers that distinguish cell types, this information is typically not exploited for scRNA-seq data integration. In this work we propose a user-friendly tool that can take advantage of available prior information on expected cell types to obtain accurate data

integration. In particular, we showed that even partial and imperfect annotations can be beneficial towards preserving biological variance while correcting for technical batch effects in integrated single-cell datasets.

In practice, some degree of prior knowledge on the cell type composition of biological samples is nearly always available. Marker genes from literature, as well as genes encoding markers commonly used in flow cytometry and immunohistochemistry are often available for many cell types. It is common practice in single-cell analyses to examine the expression of such markers to gain insights into cell cluster identity and provisionally identify cell types. We have previously shown that this task can be automated by computational tools such as scGate[33]. While complete and fine-grained annotation of cell types is challenging to achieve based only on a few marker genes, our results suggest that even incomplete prior knowledge is beneficial to data integration. Moreover, partial annotations can be complemented and refined by preliminary integration and annotation steps, suggesting a general strategy to iteratively update cell type labels and achieve improved dataset integration.

A recent benchmark[14] showed that two of the top performing methods for scRNA-seq integration were those that can make use of cell type labels information, scANVI[6] and scGen[42]. However, the performance of these methods may have been overestimated by using the same cell type labels as input for supervised integration and for evaluation of integration performance. In particular, we observed that

when introducing noisy or incomplete cell labels as input, the integration performance of scGen dropped significantly. In contrast, semi-supervised STACAS and scANVI were robust to noisy and incomplete annotations, maintaining high performance with levels of uncertainty commonly expected from single-cell datasets. We argue that benchmarks that include supervised integration methods should account for noisy and partial cell-type information, as this better reflects integration scenarios encountered in practice.

Quantifying batch effects in single-cell data is essential to assess integration quality and determine which integration methods and configurations work best in different scenarios. Commonly used batch-mixing metrics, such as LISI or Shannon entropy, are informative when evaluating batch effects between technical replicates, e.g. when there is no biological variability between samples. When integrating technical replicates, lower batch mixing between two technical replicates is a direct measurement of higher batch effects. However, most integration tasks with practical relevance involve distinct biological samples that display variation in cell type composition. In this case, a lower batch mixing does not necessarily imply higher batch effects. We argue that batch mixing metrics that neglect cell type information can overestimate batch effects between samples with large biological variance and underestimate batch effects in "overcorrected" data. This is particularly important when integrating datasets with significant cell type imbalance: in the absence of batch effects, cell type-agnostic batch-mixing metrics can increase when biological variation is removed, as cells of different type and batch are brought together as a result of batch correction algorithms. In the context of benchmarks for integration methods, this translates into favoring single-cell integration methods that "overcorrect" and penalizing methods that preserve biological variance.

To overcome this issue we propose to use cell type-aware batch mixing metrics, such as CiLISI. Because CiLISI measures batch mixing of cells of the same type only, spurious removal of cell type variance is not associated with an artificial increase in batch mixing metrics. We note that Luecken et al. [14]. were also aware of this effect and previously suggested cell type-aware modifications to existing metrics (kBET and batch ASW). One obvious limitation of CiLISI and similar metrics is that it requires cell type annotations. As discussed above, (i) the utility of cell type-agnostic batch mixing metrics is arguably very limited, and only relevant for integration of technical replicates, and (ii) it is virtually always possible to provide some level of cell type annotation. Another limitation is that CiLISI cannot be calculated for cell types that are only present in one dataset. Our implementation excludes by default these cell types from calculation. We also note that, as LISI, CiLISI remains suboptimal when datasets have highly unequal numbers of cells (e.g. even in the absence of batch effects CiLISI would be lower than the optimal value of 1 that would be observed if all cell types were equally represented). Batch mixing metrics that are more robust to lopsided cell proportions are still lacking (see ref. 43. for a systematic evaluation).

Our comprehensive benchmark using a reproducible pipeline showed that STACAS consistently ranked as the top method across multiple integration tasks and using different combinations of integration metrics. The use of prior knowledge was particularly beneficial for the integration of datasets with large cell type imbalance, where semi-supervised methods more accurately preserved relevant biological variability. On a large collection of T cell samples from cancer patients, we demonstrated the feasibility of using STACAS to integrate hundreds of samples and over 500,000 cells while preserving the diversity of T cell subtypes contained within the samples. Combining semi-supervised integration with emerging approaches to summarize massive single-cell datasets, such as metacells[44,45] and sketching[46], together with techniques for iterative 'online learning'[47], as well as on-disk, out-of-memory data representations (e.g. using DelayedArray objects[48]) will further increase scalability for organ atlas-level applications comprising millions of cells. Altogether, we propose STACAS as a first-line method for scRNA-seq data integration and encourage the broader use of prior cell type knowledge to guide integration and assess its quality.

## Methods
### STACAS integration method
STACAS is largely constructed on Seurat's anchor-based integration approach. Given as input a list of normalized expression matrices, one for each dataset, the general aim is to determine batch-effect correction vectors between pairs of datasets; and by subtracting these correction vectors to calculate a corrected data matrix of "integrated" expression values. The corrected data matrix can be used for downstream analyses such as dimensionality reduction, unsupervised clustering and cell type annotation. We will outline in the sections below the main components of the STACAS algorithm.

### Highly variable features and dimensionality reduction
The first step in STACAS integration is the calculation of highly variable genes. These are determined using the *FindVariableFeatures()* function from Seurat, but we also exclude certain classes of genes such as ribosomal, mitochondrial, and cell cycling genes that can have a large effect on the low-dimensional spaces without important contribution to cell type discrimination. These gene sets are available through the R package SignatuR (https://github.com/carmonalab/SignatuR; see e.g. https://carmonalab.github.io/STACAS.demo/STACAS.demo.html# notes-on-data-integration for an example). Additionally, genes with average log-normalized expression below (default 0.01) or above a threshold (default 3.0) are excluded from the highly variable genes. Consistently variable genes are then calculated as those found to be highly variable in multiple datasets, until reaching the desired number of genes (by default 1000). This feature selection step, also referred to as 'sharedFeatures' in the results, reduces the dimensionality of the data to a few hundred or a few thousand genes, and ensures that batch effects are calculated on genes with informative variability. The dimensionality of the data is further reduced by Principal Components Analysis (PCA) from the set of consistently variable genes. Unlike the Seurat integration method, STACAS does not by default rescale the data to zero mean and unit variance; we have previously shown how this step can mask important biological differences between datasets[49].

### Calculation and scoring of integration anchors
To determine integration anchors between pairs of datasets, we extended the reciprocal PCA algorithm (rPCA) implemented in Seurat to find shared nearest neighbors and return the pairwise distance between anchors in rPCA space (rPCA_distance). These distances are used to weigh anchor contributions, in combination with Seurat's shared nearest neighbor score (SNN_score) that quantifies the consistency of edges between cells in the same neighborhood of the SNN graph, using a geometric weighted sum with equation:

$$\text{logsum} = \alpha \ln(\text{rPCA\_score} + \varepsilon) + (1-\alpha) \ln(\text{SNN\_score} + \varepsilon) \quad (1)$$

$$\text{anchor.score} = e^{\text{logsum}} - \varepsilon \quad (2)$$

where $\varepsilon$ is a small number ($10^{-6}$) to avoid $\ln(0)$, and the rPCA distance is transformed to a score bound between 0 and 1 using:

$$\text{rPCA\_score} = logistic(\text{rPCA\_distance}) \quad (3)$$

This procedure results in a set of integration anchors with associated weights, which can be used to estimate batch effects between pairs of datasets. The parameter $\alpha$ balances the contribution of the two

scores, with α = 0.8 by default. The score weighting scheme differs from previous versions of STACAS, where suboptimal anchors were directly filtered out based on the rPCA distance scores[49]. Assigning numerical weights to anchors instead of filtering them out is more robust to parameter choices and avoids break cases where an insufficient number of integration anchors is retained after filtering, especially in the case of small datasets.

## Using prior knowledge to filter anchors

When available, prior information on cell types can be used to guide the integration by penalizing anchors composed of cells with inconsistent cell type labels. Cell type labels must be provided as a metadata column for each input object, and they can be incomplete, i.e. not all cells are required to have a label. Given a set of anchors calculated as described above and a set of cell-type labels, the algorithm rejects (with probability = *label.confidence*) anchors composed of two cells with inconsistent labels. Cells without an annotation are never rejected at this step, generalizing to the unsupervised integration scenario when no labels are available. We recommend to only provide cell type labels for high-confidence associations to a cell type, and to leave the remaining cells as unlabeled (*NA* values). The outcome of this step is a subset of the previously calculated set of integration anchors, where anchors with inconsistent cell types have been removed.

## Integration guide trees

A crucial factor in the success of batch effect correction is the order of dataset integration. To this end, STACAS calculates a weight matrix that summarizes dataset-dataset similarity. For each pair of datasets, a similarity score is obtained by summing the combined anchor scores between the two datasets. On this similarity matrix, an integration tree is calculated by applying any of the clustering methods implemented in *hclust* from the *stats* package (by default '*ward.D2*'). Integration is initiated from the dataset with the highest sum of anchor scores against all other datasets; the rationale is that the dataset with the largest number of high-scoring anchors should be the most "central", with well-represented cell types, and a large number of cells. We note that, instead, integration trees calculated by Seurat are rooted by the largest datasets, regardless of the anchor scores.

## Performance metrics of data integration

The average silhouette width (ASW) quantifies the average distance of cells in a cluster compared to their distance to the closest of the other clusters. When applied to cell type labels (celltype ASW), it measures how well cells with the same cell type label are clustered together compared to other cell types in the dataset. We compute the cell type ASW with the 'cluster' package[50] using Euclidean distances in PC space, excluding cells with unknown cell type labels.

The Local Inverse Simpson Index (LISI) has been previously presented as a metric to quantify local batch mixing (iLISI) and cell type mixing (cLISI)[11]. Briefly, LISI metrics quantify the expected number of cells from different batches (or different subtypes) in a local neighborhood, with size determined by the perplexity parameter. We propose a modified version of iLISI that does not unfairly favor overcorrection by calculating it independently for each cell type. The new cell type-aware metric, called CiLISI, is then rescaled between 0 and 1 to make it comparable across integration tasks. In all experiments in this study, we used a perplexity value of 30. Similarly, we redefined cLISI to vary between 0 and 1, where zero represents a random mix of cell types in all neighborhoods and one a perfect segregation of cell types; we call this quantity 'normalized cLISI'. For normalized cLISI we set a perplexity value to twice the average number of cells per cell type per dataset.

We implemented these metrics and made them available as an R package at: https://github.com/carmonalab/scIntegrationMetrics.

## Comparison to other integration tools

Taking advantage of the previously published 'scib' pipeline for single cell integration benchmark pipeline[14] we compared STACAS in unsupervised mode and semi-supervised mode (ssSTACAS) with 9 other integration tools: Combat[51], Scanorama[4], FastMNN[3], Harmony[11], Seurat v4 CCA and Seurat v4 rPCA[5], scVI[52], scANVI[6] and scGen[42]. We conducted our benchmark on 4 integration tasks, as detailed next.

**Integration tasks.** The human pancreas atlas, the human immune cell atlas and the human lung atlas, collated by Luecken et al. [14]., were downloaded from figshare (https://doi.org/10.6084/m9.figshare.12420968.v8). We also included in our benchmark the mouse T cell atlas by Andreatta et al. [21]., available from figshare (https://doi.org/10.6084/m9.figshare.12478571). For these 4 collections of datasets, the preprocessed data (low quality cell filtered, raw and log-normalized counts as well as original cell type annotations) obtained from the previous studies were directly analysed with the 'scib' pipeline.

**Integration procedure.** We performed the 4 integration tasks as follows:
- We used for all methods the same latent space dimensionality (D) for integration (e.g. number of principal components or dimensions of the reduced space or number of neurons in the bottleneck layer of autoencoders) and report here results for $D = 30$ and $D = 50$, the two most commonly used values in practice.
- For Seurat-based methods (rPCA and CCA), we computed reduced dimensionalities for the integrated space directly in R, starting from the scaled corrected counts matrix and applying the *RunPCA()* Seurat function; the result is used as embedding output for the 'scib' pipeline.
- All other methods were run with their default implementations as in the original benchmark; (ss)STACAS was run with the one-liner *RunSTACAS()* function and evaluated on the integrated PCA space it outputs.
- For each integration task, we used the 2000 most variable genes across the different batches (automatically identified by the 'scib' pipeline) as integration features for all methods.
- For each method, integration was performed both with or without a prior batch-aware scaling of the integration features (scaling + or − in the result summary), as implemented in the 'scib' pipeline.
- Supervised integration with scANVI, scGEN and STACAS were conducted using noisy input cell type labels. First 15% of the original cell labels were set to 'unknown', then 20% of the remaining labels were shuffled randomly.

**Integration metrics and ranking.** We first compared the methods for their ability to remove batch effect while keeping distinct cell types separated using the CiLISI and celltype_ASW, respectively. We used the original cell type labels (i.e. the "true" labels without noise) as ground truth to compute these metrics. Using the metric aggregation procedure by Luecken et al., we computed for each tool a combined score where CiLISI and cell type silhouette contribute with 60% and 40%, respectively. We calculated the mean score across all tasks to obtain a global score for each tool, which was used to compile a global ranking of the tools. Additionally, we also employed an alternative scoring scheme for tools, combining several metrics from the 'scib' pipeline: PCR batch, Batch ASW, graph iLISI, graph connectivity and kBET for the batch mixing score; NMI/cluster label, ARI cluster label, cell type silhouette, isolated label F1, isolated label silhouette, graph cLISI, cell cycle conservation and trajectory conservation (only for the immune cell atlas) for the bio-conservation score. We chose not to include HVG overlap as it can only be computed on the corrected matrix output, which is not available for all methods. As in the Luecken et al. benchmark, aggregation of metrics and ranking of tools was done by

combining bio-conservation metrics and batch mixing metrics with a relative weight of 60% and 40%, respectively. Python methods producing a corrected matrix and a corrected integrated space were evaluated on both outputs, as in the original pipeline. R methods were only evaluated on their returned corrected integrated space (Harmony, Fastmnn) or those computed in R after scaling of their returned corrected feature matrix, as intended by the developers (Seurat, STACAS).

**Robustness to noise of supervised methods.** Supervised methods (scGen, ssSTACAS, scANVI) were further benchmarked using our pipeline by providing as input shuffled or unknown cell type annotation, with increasing levels of wrong/missing annotations. Shuffled labels were generated in two alternative ways: i) by random shuffling between any pairs of cell type labels; ii) by shuffling labels between neighboring cell types, i.e. pairs of cell types with the most similar gene expression profile. In each scenario, the performance metrics (CiLISI and celltype_ASW) were evaluated on the "true" cell type labels using the top 50 principal components. We note that a similar strategy was applied to assess the impact of incorrect training labels on the performance of transfer learning across single-cell modalities[53].

**UMAP of the integrated spaces.** We calculated the two-dimensional UMAP for each method output (integrated space or corrected feature matrix) using the 'scib' pipeline. For corrected feature matrix outputs, a D-dimensional reduced space ($D = 30$ or $D = 50$) was calculated using a PCA of the corrected feature matrix, subsequently reduced to 2 dimensions using the UMAP approximation. For methods directly producing a D-dimensional integrated space, this was used directly as an input to calculate the UMAP representation.

### scGate prediction models

The scGate package[33] allows defining marker-based models for the annotation of cell types in single-cell datasets. For the annotation of human CD8 T cell subtypes, we used the collection of CD8_TIL models found at the scGate_models repository, (https://github.com/carmonalab/scGate_models, version v0.11). Briefly, the following literature-based signatures were used as gates to define subtypes: Naive-like cells: *LEF1+CCR7 + TCF7+SELL+TOX- CXCL13-*; Effector-memory: *GZMK+CXCR3+; TEMRA: FCGR3A+CX3CR1+FGFBP2+*; Precursor-exhausted: *XCL1+XCL2+TOX+GNG4+CD200+*; Terminally exhausted: *TOX+PDCD1+LAG3+TIGIT+HAVCR2+*; MAIT: *TRAV1-2+SLC4A10+*. Please refer to the models repository above for the complete combination of gates.

### Human CD8 TIL datasets

T cell scRNA-seq data were obtained from the 'utility' dataset collection (https://github.com/ncborcherding/utility), which collates data and harmonizes metadata from multiple studies and cancer types. Individual datasets were pre-processed using standard quality control, and homogenizing gene symbols according to Ensembl version 105. After filtering pure CD8 T cells, we applied UCell[54] to remove cycling cells (UCell score > 0.1) and outliers in terms of interferon response (UCell score > 0.25). We applied the CD8_TIL scGate models (see section above) to obtain a preliminary annotation of subtypes in each dataset and estimate subtype diversity. Based on these annotations, we selected 20 "seed" datasets with a large number of cells and high subtype diversity, and used them for the construction of the reference map. For all versions of STACAS, we calculated 800 variable features, and further reduced the dimensionality of the data to 50 principal components. All other parameters were used as default values. The remaining samples in the 'utility' collection were then sequentially integrated using the default STACAS pipeline, specifying the "seed" integrated map as the base dataset using the *reference* parameter. Cell type labels were transferred from the annotated "seed" map to all

remaining cells by k-nearest neighbor similarity using the *annotate.by.neighbors*() function implemented in STACAS. scANVI[6] integration on the 20 seed datasets was performed using the parameters recommended by scvi-tools, except for the dimensionality of the latent space which was set to 50 to match the number of PC dimensions used by STACAS. LIGER[47] was run in online mode by storing individual samples in.h5 files, and then executing 'online_iNMF()' and quantile_norm() on the list of.h5 files, as indicated in LIGER tutorials. We used default parameters except for k = 50 to match the dimensionalities used by STACAS and scANVI. Reproducible code for these experiments can be found at: https://github.com/carmonalab/CD8_human_TIL_atlas_construction

### Synthetic data

We generated five synthetic datasets with different levels of batch effect using the Splatter package, which applies a gamma-Poisson model to simulate gene expression distributions resembling real single-cell transcriptomics data[55]. Each dataset was composed of 1000 cells, and consisted of three batches and two cell types. The 'batch0' dataset had the three subtypes in equal proportions and zero batch effects (*batch.facLoc=0* in splatSimulate); 'batchMild' was generated by specifying *batch.facLoc=0.06* and *batch.facScale=0.06*; 'batchStrong' was generated by specifying *batch.facLoc=0.10* and *batch.facScale = 0.10*; the 'overcorrected' dataset was simulated to have no differentially expressed genes between cell types by setting *de.facLoc=0* and *de.facScale=0*. In all datasets the two cell types were set to have equal proportions.

### Statistics & Reproducibility

No statistical method was used to predetermine sample size. No data were excluded from the analyses. The experiments were not randomized. The Investigators were not blinded to allocation during experiments and outcome assessment.

### Reporting summary

Further information on research design is available in the Nature Portfolio Reporting Summary linked to this article.

## Data availability

All relevant data supporting the key findings of this study are available within the article and its Supplementary Information files. For benchmarking, the human pancreas atlas, the human immune cell atlas and the human lung atlas, collated by Luecken et al. [14]., were downloaded from figshare (https://doi.org/10.6084/m9.figshare.12420968.v8)[56]. The mouse T cell atlas by Andreatta et al. [21]. is available from figshare (https://doi.org/10.6084/m9.figshare.12478571)[57]. The T cell processed scRNA-seq dataset collection ("uTILity") used in this study was downloaded from https://doi.org/10.5281/zenodo.6325603[58]. References to the original studies and associated metadata are also available at https://github.com/ncborcherding/utility. The integrated CD8 T cell reference map is available as a Seurat object at https://doi.org/10.6084/m9.figshare.23608308[59]. Source data are provided with this paper.

## Code availability

STACAS is available as a R package at https://github.com/carmonalab/STACAS and https://doi.org/10.5281/zenodo.10402054[60]. The implementation of the performance metrics used in this work can be installed as a package from the following repository: https://github.com/carmonalab/scIntegrationMetrics also available at https://doi.org/10.5281/zenodo.10402131[61]. The snakemake pipeline that reproduces the results of our benchmark, based on the 'scib' pipeline by Luecken et al., is available at https://github.com/carmonalab/scib-pipeline and https://doi.org/10.5281/zenodo.10402023[62]. The code to construct, annotate and use the CD8 T cell reference map is available at: https://github.com/carmonalab/CD8_human_TIL_atlas_construction.

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

## Acknowledgements

We thank Nicholas Borcherding for compiling and maintaining the 'Utility' collection of tumor-infiltrating lymphocyte single-cell experiments with TCR sequencing data that were used in this study. This work was supported by the Swiss National Science Foundation (SNF) Ambizione (180010 to SJC), Swiss Cancer Research foundation (KFS-5409-08-2021 to SJC), and the ISREC foundation (to SJC).

## Author contributions

Conceptualization (M.A., S.J.C.), Methodology and Software (M.A., A.J.B., S.J.C.), Benchmarking (L.H., M.A.), Data curation (M.A., P.G.), Writing – original draft (M.A., S.J.C.), Writing – review and editing (M.A., L.H., D.G., S.J.C.), Supervision and Funding acquisition (S.J.C.).

## Competing interests

The authors declare no competing interests.
