## [Peer Review File · Nature Communications]

Semi-supervised integration of single-cell transcriptomics dataReviewer #1 (Remarks to the Author):

In this manuscript, Andreatta et al. introduce a semi-supervised integration of scRNA-seq data named STACAS. They highlight that the proposed semi-supervised framework can improve the data integration performance from the currently existing methods. The method is comprehensively evaluated with justified evaluation metrics. My main comments are detailed in the following.

Methods and findings

1. It is unclear to me what the algorithm's key and distinguished innovation is. The algorithm is based on the following components:

- (a) Identification of anchors between batches, this is mainly applying Seurat
- (b) The semi-supervised component refers to using the prior information to refine the anchors of Seurat.
- (c) Use Seurat again to perform the integration.

This method is essentially an adaptation from Seurat. However, the author has already published a paper on STACAS (Andreatta et al.), which is also an algorithm that filters on Seurat anchors. The semi-supervised component proposed in this study is built based on STACAS. To me, this is a small improvement compared to the previous version. However, the semi-supervised components have also been introduced in some other published papers, including scANVI and scSemiAE (Dong et al.). It would be great if the authors could elaborate more clearly on the innovation of their method compared to their previous version and how significant improvement of the new component has been brought by ssSTACAS compared to STACAS and Seurat.

2. Figure 3G shows scANVI performs better than ssATACAS in 3 out of 4 datasets apart from the Pancreas data which therefore makes its rankings lower than the proposed ssSTACAS, while ssSTACAS is also not ranking in the top 3 in Pancreas. Can the author elaborate on the possible reasons that scANVI is not performing well in Pancreas? Are the overall rankings shown in Figure 3G reasonable given that scANVI performs better than ssATACAS in 3 out of 4 datasets?

3. Following the last point, Figure 3 A-D and Figure 3G are actually based on different metrics, which is very confusing. If the authors think what they proposed in Figure 2 are the more reasonable metrics than Luecken et al., should they consider using the same set of metrics in Figure 3G?

4. For the sensitivity analysis in Figure 4, how would these methods perform if one label is systematically wrongly assigned to another label?

5. For Figure 6, what's the reason that the authors only benchmark with Harmony in this case study? This is a small dataset and the authors should compare their methods with all methods shown in Figure 3.

6. For Figure 7, sequential integration is an interesting case study. How will this performance compared to the other single-cell online learning data integration algorithms, such as Liger (Gao et al.) and RPCI (Liu et al.).

7. Can ssATACAS deal with the scenarios when cell type annotation is only available for one of the batches and at the same time there are novel cell types in the batch without annotations?

Minor:

1. It seems that Figure 5 is missing in the MS.

Code and software

STACAS: <https://github.com/carmonalab/STACAS>: I am able to run the tool successfully after "library(remotes)" is added in the examples in README. The codes and tutorials are well documented. A minor suggestion to maintain the usability in the long term of the software is to submit this to Bioconductor.

scIntegrationMetrics: <https://github.com/carmonalab/scIntegrationMetrics>: There is no tutorial on how to run this package.

Reference

Dong, J., Zhang, Y. & Wang, F. scSemiAE: a deep model with semi-supervised learning for single-cell transcriptomics. *BMC Bioinformatics* 23, 161 (2022). <https://doi.org/10.1186/s12859-022-04703-0>

Gao, C., Liu, J., Kriebel, A.R. et al. Iterative single-cell multi-omic integration using online learning. *Nat Biotechnol* 39, 1000–1007 (2021). <https://doi.org/10.1038/s41587-021-00867-x>

Liu, Y., Wang, T., Zhou, B. et al. Robust integration of multiple single-cell RNA sequencing datasets using a single reference space. *Nat Biotechnol* 39, 877–884 (2021). <https://doi.org/10.1038/s41587-021-00859-x>

Reviewer #2 (Remarks to the Author):

The authors have introduced a semi-supervised single-cell RNA-seq integration method driven by cell-type annotation. Throughout the manuscript, several noteworthy aspects are highlighted:

1. The authors evaluated various metrics traditionally used for benchmarking integration algorithms, ultimately deciding to use cell-type ASW and per-cell type normalized batch LISI for benchmarking.
2. A comprehensive benchmark was conducted that considers prevalent methods, considering factors such as cell type imbalance and label quality. The authors deduced that utilizing prior cell type information enhances integration, and the quality of this information is paramount.
3. They developed a strategy for iterative enhancement of cell type annotation, which in turn optimizes integration performance. This was exemplified by constructing a transcriptional map of human CD8 T Cells.
4. The capability of STACAS for extensive integration using a sequential integration strategy was demonstrated.

Nevertheless, we have identified several major and minor concerns related to the manuscript's specifics.

Major concerns:

1. In the section titled "Semi-supervised STACAS outperforms state-of-the-art methods":

- In Figure 3B and C, Seurat RPCA's performance is only slightly inferior to the supervised and semi-supervised methods.
- Figure 3D suggests that none of the methods are particularly effective on the T cell datasets, given that the average ASW is below 0.1. When combined with observations from Figure 3F, the difference between unsupervised and supervised/semi-supervised approaches doesn't appear significant.

We suggest that the author strengthens their argument by further investigating the effect of cell type imbalance on different methods. For instance, a benchmark could be conducted on the pancreas or another integration task while artificially adjusting cell type proportions in each dataset to simulate varying degrees of imbalance.

2. In the section "Semi-supervised STACAS is robust to incomplete and noisy annotations", the authors didn't show batch mixing metrics results from both the main text and supplementary figures. A comprehensive evaluation should include batch mixing performance across all methods.

3. In the third paragraph of the same section, references to main benchmark results seem to be absent from both the manuscript and the accompanying figures.

Minor concerns:

1. It would be beneficial if the authors could present UMAP plots labeled by batches for all benchmarked methods during the T cell datasets integration task.
2. The shuffling and unknown label settings appear to be overly stringent. In practical scenarios, cell identity ambiguity typically arises at margins between clusters. The rationale for this experimental design should be clarified. Is it universally applicable and fair across all methods?
3. If the author could kindly provide the information about computational time and resources needed in the large-scale integration example, it would help readers prepare their own practice in using STACAS.

Response to reviewers

Manuscript NCOMMS-23-33775-T – “Semi-supervised integration of single-cell transcriptomics data” by Andreatta et al.

We would like to thank the reviewers for taking the time to evaluate our manuscript and for providing useful comments and suggestions. In the following, we address all reviewer comments and describe updates to the manuscript. Answers to reviewers in this document and edits in the revised manuscript are highlighted in red. We hope you will find the revised manuscript suitable for publication in Nature Communications.

Reviewer #1 (Remarks to the Author):

In this manuscript, Andreatta et al. introduce a semi-supervised integration of scRNA-seq data named STACAS. They highlight that the proposed semi-supervised framework can improve the data integration performance from the currently existing methods. The method is comprehensively evaluated with justified evaluation metrics. My main comments are detailed in the following.

Methods and findings

1. It is unclear to me what the algorithm's key and distinguished innovation is. The algorithm is based on the following components:

- (a) Identification of anchors between batches, this is mainly applying Seurat
- (b) The semi-supervised component refers to using the prior information to refine the anchors of Seurat.
- (c) Use Seurat again to perform the integration.

This method is essentially an adaptation from Seurat. However, the author has already published a paper on STACAS (Andreatta et al.), which is also an algorithm that filters on Seurat anchors. The semi-supervised component proposed in this study is built based on STACAS. To me, this is a small improvement compared to the previous version. However, the semi-supervised components have also been introduced in some other published papers, including scANVI and scSemiAE (Dong et al.). It would be great if the authors could elaborate more clearly on the innovation of their method compared to their previous version and how significant improvement of the new component has been brought by ssSTACAS compared to STACAS and Seurat.

Thank you for giving us the opportunity to better explain the innovations introduced by ssSTACAS and by this study.

With regards to the algorithm, ssSTACAS differs from STACAS-1.0 in several aspects. First, we implemented a new strategy for anchor weighting (described in the section “Calculation and scoring of integration anchors”) that differs from previous versions of STACAS, where anchors were directly filtered out based on rPCA scores. Such re-weighting is more robust to parameter choices and avoids break cases where an insufficient number of integration anchors is retained after filtering, especially in the case of small datasets (as also previously reported by Richards et al. 2021). Second, ssSTACAS implements an improved scheme for determining the order of sample integration (section “Integration guide trees”), which is based on the sum of anchor scores between pairs of datasets. Third, ssSTACAS now includes a strategy for sequential integration, whereby full pairwise integration is performed on a reduced number of “seed” datasets, and all other datasets are then added sequentially to the “seed” map. This strategy allows for large scale integrations, as exemplified in the integration of 265 samples covering >500,000 cells (Figure 6). Fourth, ssSTACAS enables the use of prior knowledge to guide integration. While this idea has been proposed before (e.g. scANVI and scSemiAE), it represents a significant innovation compared to STACAS and Seurat-based methods, and it is shown to outperform existing methods.

Besides algorithmic improvements, the manuscript brings several additional novel contributions. First, we evaluated multiple metrics for assessing integration performance, and proposed two metrics that were shown to be particularly informative (CiLISI and celltype_ASW). These and other metrics were made available to the community as an R package. Second, we conducted a comprehensive benchmark of integration tools across multiple tasks, considering factors such as cell type imbalance and cell type label quality. We showed that prior cell type information is beneficial

for integration, even when incomplete, and we advocate for wider adoption of semi-supervised integration. Third, we assessed the robustness of semi-supervised methods to noisy and missing annotations, as are commonly encountered in integration tasks. To the best of our knowledge, this kind of analysis has not been previously reported. This is critical because previous benchmarks did not account for missing and noisy annotations and thus overestimated the performance of supervised methods. We provide a fully reproducible benchmark pipeline that includes these new criteria. Fourth, we described a strategy for iteratively updating cell type prior knowledge, leading to improved integration results, as exemplified by the construction of a multi-sample reference map of human CD8 T cells (**Figure 5**).

Taken all together, we believe these novel contributions are worth reporting in a published manuscript. We have updated the manuscript to highlight the points listed above in several points of the Methods and the Discussion.

2. Figure 3G shows scANVI performs better than ssATACAS in 3 out of 4 datasets apart from the Pancreas data which therefore makes its rankings lower than the proposed ssSTACAS, while ssSTACAS is also not ranking in the top 3 in Pancreas. Can the author elaborate on the possible reasons that scANVI is not performing well in Pancreas? Are the overall rankings shown in Figure 3G reasonable given that scANVI performs better than ssATACAS in 3 out of 4 datasets?

Thank you for bringing up this point, which will allow us to better explain the relative ranking of the methods. When evaluating the overall score to determine global rankings, the performance of ssSTACAS and scANVI is essentially identical in 2 of the 4 tasks; scANVI outperforms ssSTACAS on the T cell task; and ssSTACAS outperforms scANVI on the Pancreas task (**Figure 3G** and **Reviewer Fig. 1A** shown below). Because the global rankings are calculated as the average of the scores (and not the ranks) across tasks, and the difference is larger in the Pancreas task than in the T cell task, ssSTACAS comes out on top (**Figure 3G**).

As for the possible reasons why scANVI performs poorly on the Pancreas dataset, we looked into the rankings for individual metrics for the Pancreas integration task (**Reviewer Fig. 1 B**). scANVI appears to perform suboptimally for several metrics, in particular kBET, cell type ASW, graph cLISI and CC conservation. We cannot pinpoint the exact reasons for this poor performance, but it involves both suboptimal batch mixing and poor bio-conservation.

Notably, when evaluated using our recommended metrics CiLISI and celltype_ASW, instead of the metrics proposed by Luecken et al, scANVI drops to 8th/9th position in the rankings, whereas ssSTACAS remains the best performing method (**Figure S3B,D**). We briefly discuss these aspects in the revised manuscript (lines 211-218).

Reviewer Figure 1: A) Combined overall score, batch correction score and bio-conservation score for semi-supervised STACAS, scANVI and Unintegrated data over 4 integration tasks. Boxplot in black represents distribution of scores for the remaining tools (minimum, first quartile, median, third quartile, and maximum). **B)** Overall ranking and individual metrics for the Pancreas integration task, calculated using the scib pipeline.

3. Following the last point, Figure 3 A-D and Figure 3G are actually based on different metrics, which is very confusing. If the authors think what they proposed in Figure 2 are the more reasonable metrics than Luecken et al., should they consider using the same set of metrics in Figure 3G?

Thank you for the suggestion. We do think that CiLISI and celltype_ASW are the best metrics to evaluate batch mixing and preservation of biological variability, respectively. We display benchmark results in terms of these metrics in scatterplots in **Figure 3A-D** and calculate global rankings based on these metrics in **Suppl. Fig S3 B,D**. To complement these rankings, we also calculated performance rankings based on the larger set of metrics proposed by Luecken et al., which are currently displayed in **Figure 3G**.

We chose this presentation of results to avoid giving the impression that the metrics CiLISI and celltype_ASW were selected to favor our method, and to complement **Figure 3A-D** with additional, composite metrics. Regardless of the metrics used (our metrics, or the combined metrics) ssSTACAS ranks as the best-performing method. We have now added a subtitle to **Figure 3G** to highlight that the rankings were calculated on multiple integration metrics. If the reviewers and editor still believe this presentation of results is confusing, we are open to swapping **Figure 3G** with **Figure S3D**.

4. For the sensitivity analysis in Figure 4, how would these methods perform if one label is systematically wrongly assigned to another label?

This is an interesting point that was also suggested by Reviewer 2 (minor concern 2). We conducted a new robustness analysis in which we systematically mixed the labels of each cell type with the labels of its transcriptionally closest cell type, and evaluated integration performance. This analysis shows that performance of ssSTACAS and scANVI is maintained relatively constant with increasing levels of noise, while performance of scGen decays abruptly, also when cell type missannotation affects only neighboring cell types. This confirmed the previous results using completely random cell type mixing, demonstrating that ssSTACAS is robust to wrong cell type labels and leads to increased preservation of biological variation. This experiment is presented in **Figure S7** and described in the updated manuscript (lines 265-267 and Methods).

Figure S7: Effect of noisy cell type annotations (by nearest neighbor) on data integration by supervised or semi-supervised methods. Preservation of biological variance (measured by celltype_ASW) and batch mixing (measured by CiLISI) for 4 data integration tasks, using as input all correct cell type labels (0%) or increasing levels of shuffled cell type labels (5% to 50%). In these experiments, labels were shuffled between neighboring cell types, i.e. pairs of cell types with the most similar average expression profiles. Note that shuffling is bounded to 50% of the labels, as higher values would start increasing the probability of cells of the same type being assigned to the same label.

5. For Figure 6, what's the reason that the authors only benchmark with Harmony in this case study? This is a small dataset and the authors should compare their methods with all methods shown in Figure 3.

We conducted a comprehensive benchmark using an open, reproducible pipeline in the section “Semi-supervised STACAS outperforms state-of-the-art methods”. The purpose of the analysis shown in Figure 6 (now **Figure 5**) is not to repeat the benchmark, but rather to showcase the application of STACAS to a specific integration task, and in particular highlight our novel strategy of iterative update of prior knowledge.

We believe that **Figures 3** and **4**, and Supplementary **Figures S3, S4, S5, S6** and **S7** already provide a comprehensive overview of the benchmark results against multiple methods within a consistent evaluation framework and independent ground truth labels. We included specifically Harmony in **Figure 5** to provide an approximate baseline of performance using what is arguably the most popular method for single-cell integration.

6. For Figure 7, sequential integration is an interesting case study. How will this performance compared to the other single-cell online learning data integration algorithms, such as Liger (Gao et al.) and RPCI (Liu et al.).

As discussed in our previous answer, method benchmarking was covered in previous sections of the manuscript (in particular in **Figures 3** and **4**, and Supplementary **Figures S3, S4, S5, S6** and **S7**). The goal of this analysis is to show the scalability of our method and highlight its sequential integration strategy.

For reference, LIGER was shown to be one of the worst-performing methods for scRNA-seq integration in the original benchmark by Luecken et al. (cfr. Figure 3 of their paper: <https://www.nature.com/articles/s41592-021-01336-8/figures/3>). RPCI was not included in the benchmark by Luecken et al. for not satisfying the inclusion criteria, which consider among other factors regular maintenance and solving open issues in GitHub. We added references to Liu et al. in the Introduction (line 47) and to Gao et al. in the Discussion (line 439).

7. Can ssATACAS deal with the scenarios when cell type annotation is only available for one of the batches and at the same time there are novel cell types in the batch without annotations?

Thank you for bringing up this important point. First, we would like to clarify that when cell type annotations are available for only one of the batches, ssSTACAS will behave in unsupervised mode. This is because unlabeled cells are treated as wildcards and are freely allowed to form anchor pairs with any cell. Therefore, in order to benefit from prior information, cell type labels must be available for more than one batch.

As for the problem of having an unannotated “novel” cell type (i.e. an unknown cell type that is present in only one sample), this is central to batch correction. In fact, it can be considered a specific case of the cell type imbalance problem, which we have thoroughly addressed in our benchmark. In particular, we have shown that in a task with large cell type imbalance, both semi-supervised and unsupervised STACAS showed the highest preservation of biological variability with a good level of batch mixing (T cell task, **Figure 3 D**). This means that even in the absence of annotations, STACAS is able to integrate without mixing sample-specific (“novel”) cell types (e.g. CD4_NaiveLike; >90% cells come from one sample MC38_dLN, **Figure S1 D**).

To illustrate with a simple example how semi-supervised STACAS can handle cases where a novel, unannotated cell type is only present in one batch, we performed the following experiment. We selected two batches from the pancreas dataset (‘celseq2’ and ‘inDrop1’), and the 7 most abundant cell types. Then, we removed all acinar cells from the celseq2 sample, and set all acinar cells in the inDrop1 sample to NA. Therefore, this task includes one ‘novel’, unannotated cell type which is present only in one of the batches. Semi-supervised STACAS could successfully mitigate batch effects and bring together cells of the same type (**Reviewer Figure 2**), and at the same time cells from the novel cell type (i.e. the unlabeled acinar cells) were not mixed with other cell types. This experiment demonstrates that ssSTACAS can handle cases where there is a novel, unlabeled cell type which is present in only one batch.

Reviewer Figure 2: integration task composed of two batches (*celseq2* and *inDrop1*) and 7 cell types from pancreas. All acinar cells were removed from the *celseq2* batch, and all acinar cells from the *InDrop1* batch were labeled as NA (i.e. they were left unannotated). Upon integration, *ssSTACAS* mitigated batch effects and brought cells of the same types together, while at the same time unlabeled cells (NA, in gray) were not mixed with other cell types.

In this revised manuscript, we further explored the effect of cell type imbalance by systematically removing N cell types from individual studies/batches (with N between 0 and 7). This experiment of simulated imbalance in the (otherwise well-balanced) Pancreas dataset showed that both semi-supervised and unsupervised STACAS maintained the highest performance even when up to 7 random cell types were removed from each batch (**Figure S6**, also shown below). Of note, we stopped at 7 because removing more than 7 subtypes caused some methods (but not STACAS) to break. Please also see the response to Reviewer 2, point 1 for a more detailed description of this analysis.

Figure S6: Global ranking of integration tools on Pancreas dataset with increasing levels of cell type imbalance. Datasets 'Remove N' (with N between 0 and 7) were generated by randomly removing N cell types from each sample. The plots display combined integration scores calculated using the 'scib' pipeline, and 1,2,3 identify the top three ranking methods for each integration task. **A)** Methods ranking based on CiLISI and celltype_ASW metrics; **B)** Methods ranking based on combination of multiple integration metrics from the 'scib' pipeline.

Minor:

1. It seems that Figure 5 is missing in the MS.

Thank you for pointing this out, the figures were misnumbered. This has been fixed.

Code and software

STACAS: <https://github.com/carmonalab/STACAS>: I am able to run the tool successfully after "library(remotes)" is added in the examples in README. The codes and tutorials are well documented. A minor suggestion to maintain the usability in the long term of the software is to submit this to Bioconductor.

We are happy the reviewer could successfully run the package and the tutorials. From our experience, packages that directly depend on Seurat instead of using SingleCellExperiment infrastructure are generally not accepted by Bioconductor; however, we have started work towards submitting STACAS to CRAN.

scIntegrationMetrics: <https://github.com/carmonalab/scIntegrationMetrics>: There is no tutorial on how to run this package.

We have now included detailed usage information on the README page of the package, which is directly displayed on the landing page of the repository: <https://github.com/carmonalab/scIntegrationMetrics>

Reference

Dong, J., Zhang, Y. & Wang, F. scSemiAE: a deep model with semi-supervised learning for single-cell transcriptomics. BMC Bioinformatics 23, 161 (2022). <https://doi.org/10.1186/s12859-022-04703-0>

Gao, C., Liu, J., Kriebel, A.R. et al. Iterative single-cell multi-omic integration using online learning. Nat Biotechnol 39, 1000–1007 (2021). <https://doi.org/10.1038/s41587-021-00867-x>

Liu, Y., Wang, T., Zhou, B. et al. Robust integration of multiple single-cell RNA sequencing datasets using a single reference space. Nat Biotechnol 39, 877–884 (2021). <https://doi.org/10.1038/s41587-021-00859-x>

Thank you for these references, we have now included them in the manuscript.

Reviewer #2 (Remarks to the Author):

The authors have introduced a semi-supervised single-cell RNA-seq integration method driven by cell-type annotation. Throughout the manuscript, several noteworthy aspects are highlighted:

1. The authors evaluated various metrics traditionally used for benchmarking integration algorithms, ultimately deciding to use cell-type ASW and per-cell type normalized batch LISI for benchmarking.
2. A comprehensive benchmark was conducted that considers prevalent methods, considering factors such as cell type imbalance and label quality. The authors deduced that utilizing prior cell type information enhances integration, and the quality of this information is paramount.
3. They developed a strategy for iterative enhancement of cell type annotation, which in turn optimizes integration performance. This was exemplified by constructing a transcriptional map of human CD8 T Cells.
4. The capability of STACAS for extensive integration using a sequential integration strategy was demonstrated. Nevertheless, we have identified several major and minor concerns related to the manuscript's specifics.

Thank you for recognizing the importance of our work.

Major concerns:

1. In the section titled “Semi-supervised STACAS outperforms state-of-the-art methods”:
 - In Figure 3B and C, Seurat RPCA's performance is only slightly inferior to the supervised and semi-supervised methods.

While in certain tasks the improvement of STACAS over rPCA was not very large, STACAS was consistently better at preserving biological variability (**Figure 3 A-D, Figure S4 A-D**). Using our suggested metrics (CiLISI and celltype_ASW), rPCA ranked 5th/4th using 30 or 50 PC for dimensionality reduction, respectively (**Figure S3B,D**). When considering the larger panel of metrics, rPCA ranked 4th to 8th depending on the number of PCs. In contrast, STACAS was consistently the top performing method, regardless of metrics used or the number of PCs for dimensionality reduction (**Figure S3 A,C**). For these reasons we believe it is accurate to state that semi-supervised STACAS outperforms state-of-the-art methods, including Seurat RPCA.

- Figure 3D suggests that none of the methods are particularly effective on the T cell datasets, given that the average ASW is below 0.1. When combined with observations from Figure 3F, the difference between unsupervised and supervised/semi-supervised approaches doesn't appear significant. We suggest that the author strengthens their argument by further investigating the effect of cell type imbalance on different methods. For instance, a benchmark could be conducted on the pancreas or another integration task while artificially adjusting cell type proportions in each dataset to simulate varying degrees of imbalance.

Thank you for this excellent suggestion. We have performed the suggested experiment and artificially generated modified versions of the pancreas dataset with increasing levels of cell type imbalance. This was achieved by randomly removing N cell types from each sample (with N between 0 and 7), and then applying all tools to integrate these increasingly imbalanced datasets. The results are presented in the new **Figure S6** (also shown below), where the datasets are named 'Remove N', and the tools are ranked according to the global performance on these versions of the pancreas dataset.

From these results it is clear that ssSTACAS consistently outperforms the other methods across a wide range of cell type imbalance. In particular, when evaluating rankings based on CiLISI and celltype_ASW, ssSTACAS is the top performing method in all cases except 'Remove 0' (fully balanced set). Consistently, the other supervised methods

scANVI and scGen show a relative increase in performance (compared to other, unsupervised methods) as cell type imbalance increases. In contrast, Seurat CCA is the top performer with balanced cell type composition, but its performance drops sharply as soon as imbalance is added to the data (**Figure S6A**). A noisier but similar trend can be observed for the other unsupervised methods (e.g. Harmony, Seurat RPCA had decreasing performance as cell type imbalance increased). The rankings based on all metrics (as opposed to CiLISI and celltype ASW) show a slightly noisier and less linear behavior with respect to cell type imbalance, but confirm the same patterns (**Figure S6B**). In particular, ssSTACAS is in the top 3 methods across all levels of imbalance, and the performance of unsupervised methods tends to decrease with high levels of imbalance.

Altogether, this analysis confirmed that the relative performance of supervised methods tends to increase with the degree of cell type imbalance, compared to unsupervised methods. In particular, the performance of semi-supervised STACAS across a wide range of cell type imbalance is superior to competing methods. These results are presented in **Figure S6** and described in the updated manuscript (lines 221-232).

Figure S6: Global ranking of integration tools on Pancreas dataset with increasing levels of cell type imbalance. Datasets 'Remove N' (with N between 0 and 7) were generated by randomly removing N cell types from each sample. The plots display combined integration scores calculated using the 'scib' pipeline, and 1,2,3 identify the top three ranking methods for each integration task. **A)** Methods ranking based on CiLISI and celltype_ASW metrics; **B)** Methods ranking based on combination of multiple integration metrics from the 'scib' pipeline.

2. In the section "Semi-supervised STACAS is robust to incomplete and noisy annotations", the authors didn't show batch mixing metrics results from both the main text and supplementary figures. A comprehensive evaluation should include batch mixing performance across all methods.

We agree this information is important to provide a comprehensive view of the analysis. We have updated **Figure 4** to include batch mixing (measured by CiLISI) as a function of incorrect / missing annotations. Batch mixing results also seem to indicate that STACAS and scANVI are more robust to noisy labels than scGen. These new results are presented in **Figure 4** and discussed in the manuscript (lines 261-262).

Updated Figure 4: Effect of noisy or incomplete cell type annotations on data integration by supervised or semi-supervised methods. **A)** Preservation of biological variance (measured by *celltype_ASW*) and batch mixing (measured by *CiLISI*) for 4 data integration tasks, using as input all cell type labels (original) or increasing levels of shuffled cell type labels (10% to 100%). **B)** Preservation of biological variance (measured by *celltype_ASW*) and batch mixing (measured by *CiLISI*) for 4 data integration tasks, using as input all cell type labels (original) or increasing fractions of unknown cell type labels (10% to 100%). Unsupervised versions of *ssSTACAS* and *scANVI* (*STACAS* and *scVI* respectively) are included for reference.

3. In the third paragraph of the same section, references to main benchmark results seem to be absent from both the manuscript and the accompanying figures.

The paragraph refers to the main benchmark previously presented in **Figure 3**. We realized the original text was not clear and we have rephrased it to avoid confusion (lines 272-274):

“These considerations suggest that fair performance evaluation of semi-supervised tools should account for potential noise in the input labels. As previously mentioned, the benchmark presented in Figure 3 included 20% shuffled labels and 15% unknown labels.”

Minor concerns:

1. It would be beneficial if the authors could present UMAP plots labeled by batches for all benchmarked methods during the T cell datasets integration task.

Thank you for the suggestion. We have generated UMAP plots labeled by study, which are presented in the new **Figure S2**.

Figure S2: UMAP embeddings for the mouse T cell integration task colored by batch, before integration (Unintegrated) and after integration using eight different integration methods.

2. The shuffling and unknown label settings appear to be overly stringent. In practical scenarios, cell identity ambiguity typically arises at margins between clusters. The rationale for this experimental design should be clarified. Is it universally applicable and fair across all methods?

We believe the outlined scenarios are *i)* reasonable and plausible, because manual annotation or automated tools cannot confidently assign the totality of cells to a cell type; *ii)* universally applicable, because masking and shuffling cell labels is a straightforward task which we have implemented in the modified 'scib' pipeline; and *iii)* fair across methods, since they all receive the same input and thus are subjected to the same noise levels.

Nevertheless, we agree with the reviewer that misannotated cells may preferentially arise at margins between clusters, or assigned by prediction methods to neighboring, transcriptionally similar cell types rather than to random cell types. To evaluate this alternative scenario, we explored a different strategy to shuffle cell type labels, by systematically wrongly annotating a percentage of cells from each cell type to its most transcriptionally similar cell type. This analysis shows that performance of ssSTACAS and scANVI is maintained relatively constant with increasing levels of noise, while performance of scGen decays abruptly, also when cell type missannotation affects only neighboring cell types. This confirmed the previous results using completely random cell type mixing, demonstrating that ssSTACAS is robust to wrong cell type labels and leads to increased preservation of biological variation. These results are presented in **Figure S7** and described in the updated manuscript (lines 265-267).

Figure S7: Effect of noisy cell type annotations (by nearest neighbor) on data integration by supervised or semi-supervised methods. Preservation of biological variance (measured by celltype_ASW) and batch mixing (measured by CiLISI) for 4 data integration tasks, using as input all correct cell type labels (0%) or increasing levels of shuffled cell type labels (5% to 50%). In these experiments, labels were shuffled between neighboring cell types, i.e. pairs of cell types with the most similar average expression profiles. Note that shuffling is bounded to 50% of the labels, as higher values would start increasing the probability of cells of the same type being assigned to the same label.

3. If the author could kindly provide the information about computational time and resources needed in the large-scale integration example, it would help readers prepare their own practice in using STACAS.

We have included this information in the corresponding section of the manuscript (lines 357-358): “This operation could be completed in approximately 150 minutes on a desktop computer with 64GB of RAM.”

Reviewer #1 (Remarks to the Author):

The authors have addressed several of the comments previously made. Nonetheless, I still have two main concerns that remain unaddressed.

1. The level of novelty claimed for this method seems to be overstated. For example, they claimed, "To the best of our knowledge, this kind of analysis has not been previously reported." However, this perspective is already examined in other single-cell studies, such as scJoint (Lin et al. 2022, Nature Biotech), which is also a semi-supervised single-cell data integration paper.

2. The authors are reluctant to conduct a comprehensive benchmarking of their methods in the last two case studies.

(1) In the first case study, Harmony was used as a benchmark, which ranks 11th in Figure 3G. It would seem that using the top-ranked methods as a baseline would be more appropriate. Additionally, the performance of ssSTACAS, as shown in Figures 3-4, is very similar to methods like scANVI, which doesn't convincingly demonstrate a significant improvement over other published methods.

(2) For the last case study, they refer to Luecken et al. to exclude the methods that perform very similar tasks to the proposed methods. However, in Luecken et al., the Liger method being benchmarked is the one published in 2019 (Welch et al. 2019, Cell), not the one with an online learning version (Gao et al. 2021, Nature Biotech). On the other hand, Luecken et al. do not explicitly state the inclusion criterion mentioned by the authors. Luecken et al. only claim they included the most popular 16 methods for benchmarking, as opposed to what the authors claimed that the criterion is "regular maintenance and solving open issues in Github" in their response letter. Therefore, the argument for not including these methods in the benchmark seems poorly justified to me.

Reviewer #2 (Remarks to the Author):

All concerns are addressed.

Response to reviewers

Manuscript NCOMMS-23-33775-T – “Semi-supervised integration of single-cell transcriptomics data” by Andreatta et al.

Reviewer #1 (Remarks to the Author):

The authors have addressed several of the comments previously made. Nonetheless, I still have two main concerns that remain unaddressed.

We thank once more the reviewer for conducting a detailed evaluation of our work and for their valuable suggestions, which have helped us further strengthen our manuscript. In this revised version we have addressed the remaining concerns.

1. The level of novelty claimed for this method seems to be overstated. For example, they claimed, “To the best of our knowledge, this kind of analysis has not been previously reported.” However, this perspective is already examined in other single-cell studies, such as scJoint (Lin et al. 2022, Nature Biotech), which is also a semi-supervised single-cell data integration paper.

Thank you for pointing out this reference. In Lin et al. the authors evaluated the impact of shuffling training labels (5%, 10% or 20%) on the performance capability of their method to transfer labels from scRNA-seq data to scATAC-seq data, which is a different but related task. We have now referenced this related analysis in the updated manuscript (lines 593-595):

“We note that a similar strategy was applied to assess the impact of incorrect training labels on the performance of transfer learning across single-cell modalities⁵³.”

2. The authors are reluctant to conduct a comprehensive benchmarking of their methods in the last two case studies.

(1) In the first case study, Harmony was used as a benchmark, which ranks 11th in Figure 3G. It would seem that using the top-ranked methods as a baseline would be more appropriate. Additionally, the performance of ssSTACAS, as shown in Figures 3-4, is very similar to methods like scANVI, which doesn't convincingly demonstrate a significant improvement over other published methods.

(2) For the last case study, they refer to Luecken et al. to exclude the methods that perform very similar tasks to the proposed methods. However, in Luecken et al., the Liger method being benchmarked is the one published in 2019 (Welch et al. 2019, Cell), not the one with an online learning version (Gao et al. 2021, Nature Biotech). On the other hand, Luecken et al. do not explicitly state the inclusion criterion mentioned by the authors. Luecken et al. only claim they included the most popular 16 methods for benchmarking, as opposed to what the authors claimed that the criterion is “regular maintenance and solving open issues in Github” in their response letter. Therefore, the argument for not including these methods in the benchmark seems poorly justified to me.

In the first part of our manuscript, we conducted a comprehensive benchmark of multiple tools on several integration tasks with ground-truth cell type information (Figures 3, 4, S3, S4, S5, S6, S7). These results show that STACAS is consistently the top performing method regardless of the metrics used and of the dimensionality of the latent spaces (Figure S3); scANVI was the second-best method in some cases, but also dropped to 8th or 9th depending on the set of metrics used. Additionally, STACAS showed consistently higher cell type silhouette coefficient than scANVI in the shuffled/partial labels experiments (Figure 4). Finally, STACAS also clearly outperformed scANVI and the other methods in the integration tasks with simulated imbalance presented in Figure S6. For these reasons, we respectfully disagree that the performance of STACAS is similar to that of other methods.

After these benchmarks, in the last section of our manuscript we sought to highlight several features of our integration method on a case study. In this case study, we first generated prior cell type information through an automated marker-based classifier. Samples were integrated using semi-supervised STACAS

and the prior cell type information, to enable a first integrated space. On this integrated space, we refined cell type labels (as users may commonly do for cell type annotation) and showed that we could then restart a new integration from scratch guided by these refined cell type labels. Finally, we showed the feasibility of adding >250 additional samples to this map by STACAS sequential integration. We believe this general strategy can be of guidance to users in their integration tasks. However, we hope the reviewer will appreciate that this case study (which is not a benchmark) involves steps that are not fully automatic and that complicate direct comparisons between method; such comparisons were thoroughly performed within a reproducible benchmarking pipeline in previous sections of the manuscript. We had explicitly included Harmony integration results as a baseline in the case study for an out-of-the-box integration using arguably the most popular method currently available. In this revised version, following the reviewer's recommendation, we have now also evaluated the applicability of scANVI (the second best performing method of the benchmark) and LIGER on this case study (Fig. S8B and new Fig. S9).

Figure S8B: Batch mixing (measured by CiLISI) and bio-conservation (measured by celltype_ASW) for the indicated algorithms on the collection of CD8 T cell samples.

This new result is now referenced in the updated manuscript lines 315 and 348.

In terms of integration metrics for the 20 seed datasets of the case study, we found that scANVI had comparable performance to the first semi-supervised STACAS integration (slightly higher celltype_ASW and slightly lower CiLISI, Figure S8B), and lower celltype_ASW and CiLISI compared to the second iteration of semi-supervised STACAS integration. LIGER performed similarly to Harmony, where both had celltype_ASW that was no better than the uncorrected merging of the datasets (Figure S8B).

As the reviewer remarked, we verified that regular maintenance and Github activity were not reported by Luecken et al. as criteria for including tools in their benchmark. Instead, usability assessment, which includes “quality of the code, its availability, the presence of a tutorial to guide users through one or more examples, GitHub issue activity and responsiveness”, was only used as an evaluation criterion in the benchmark. We apologize for this inaccurate statement in our response letter.

As suggested by the reviewer, we also evaluated whether LIGER in online-learning mode could be applied for large scale integration of all 285 human CD8 T samples. We would like to remark that online learning in the context of LIGER, as described in their own manuscript (Gao et al. Nature Biotech), brings improvements in terms of speed and memory usage, but not in integration accuracy compared to the “batch” version of LIGER. In this case study, we saw that these >500,000 T cells could be integrated by LIGER, allowing us to define 7 unsupervised clusters (Figure S9A). This number of clusters was chosen to be able to compare with the 7 subtypes defined in the STACAS-integrated space (Figure 6). We observed that the unsupervised clusters in the LIGER-integrated space poorly reflected the expected

subtypes of CD8 T cells (Figure S9B). In particular, there was no cluster capturing TEMRA cells (characterized by *FGFBP2*, *FCGR3A* and *KLRG1*); no *XCL1+CRTAM+TOX+* precursor-exhausted T cells; and no cluster of MAIT cells (characterized by *SLC4A10* and by the invariant chain *TRAV1-2*). All of these and other subtypes were successfully captured in the integrated space defined by STACAS integration. These results are now included in the new Figure S9.

Figure S9: Large scale integration of CD8 T cells using LIGER. A) UMAP embeddings of 285 samples integrated using LIGER in online-learning mode, colored by unsupervised cluster (left) and by predicted scGate CD8 T cell subtypes. **B)** Expression profiles for unsupervised clusters obtained by online LIGER integration, using a panel of standard marker genes for human CD8 T cell subtypes.

This new result is now referenced in the updated manuscript lines 363 and 375. The updated methods are included in lines 628-634.

Finally, please note that we have updated the low-dimensional plots for Harmony in Supplementary Fig. S8A. We previously used the first two dimensions of the 'harmony' embeddings to display these data, which were not directly comparable to UMAP embeddings used for the other methods. We now corrected these panels to also use UMAP embeddings for Harmony. These updates do not affect the quantitative results and conclusions of these experiments.

Reviewer #1 (Remarks to the Author):

I appreciate the authors' responses with the additional experiments. The authors have addressed all the comments I have made in the previous round. I have no further comments.